# REPTOR and CREBRF encode key regulators of muscle energy metabolism

Pedro Saavedra [1] ✉, Phillip A. Dumesic [2,3], Yanhui Hu [1], Elizabeth Filine [1], Patrick Jouandin [4], Richard Binari [1,5], Sarah E. Wilensky [2], Jonathan Rodiger [1], Haiyun Wang [6], Weihang Chen [1], Ying Liu [1], Bruce M. Spiegelman [2,3] & Norbert Perrimon [1,5] ✉

Metabolic flexibility of muscle tissue describes the adaptive capacity to use different energy substrates according to their availability. The disruption of this ability associates with metabolic disease. Here, using a *Drosophila* model of systemic metabolic dysfunction triggered by yorkie-induced gut tumors, we show that the transcription factor REPTOR is an important regulator of energy metabolism in muscles. We present evidence that REPTOR is activated in muscles of adult flies with gut yorkie-tumors, where it modulates glucose metabolism. Further, in vivo studies indicate that sustained activity of REPTOR is sufficient in wildtype muscles to repress glycolysis and increase tricarboxylic acid (TCA) cycle metabolites. Consistent with the fly studies, higher levels of CREBRF, the mammalian ortholog of REPTOR, reduce glycolysis in mouse myotubes while promoting oxidative metabolism. Altogether, our results define a conserved function for REPTOR and CREBRF as key regulators of muscle energy metabolism.

Metabolic flexibility refers to the ability of tissues like the muscle to use glucose or lipids as energy substrates depending on their availability[1]. Glucose is processed to pyruvate in the presence of oxygen in the cytoplasm by a cascade of sequential enzymatic steps that constitute the glycolysis pathway; lipids are stored in the form of triglycerides in lipid droplets of the adipose tissue cells, but during lipolysis, fatty acids are released into circulation to be used by the mitochondria of muscle cells, where they undergo beta-oxidation. Pyruvate and fatty acids are then used as substrates to produce Acetyl-CoA, which is incorporated into the TCA cycle in mitochondria to stimulate ATP production[2,3].

Both glucose and lipids are used interchangeably in muscle as oxidative fuel, as an adaption to the nutritional status[4]. After feeding, an increase in glucose levels reduces fatty acid oxidation in muscle, whereas nutritional restriction suppresses glucose oxidation while increasing the breakdown of triglycerides and supply of fatty acids to the muscle mitochondria[5,6]. Disruption of this flexibility, however, is frequently observed in the context of overwhelming abundance of either glucose or lipids and associates with insulin resistance, obesity and disruption of muscle energy metabolism[7–10]. It is therefore important to identify genes that regulate energy substrate use by muscle tissue, as they may have therapeutic potential in the context of metabolic disease.

Cancer induced-cachexia is a systemic metabolic dysfunction observed in cancer patients that severely affects skeletal muscle and adipose tissue[11–13]. During cachexia, skeletal muscle displays impaired glucose metabolism and insulin resistance[14], whereas the adipose tissue often exhibits higher levels of lipolysis prior to muscle wasting[15,16]. Likewise, changes in glucose/fatty acid utilization are frequently observed in muscle of cachexic patients[17]. Furthermore, alterations in mitochondrial function in muscle have also been associated with wasting and loss of sarcomere integrity during cachexia[18,19], suggesting a link between energy metabolism and muscle wasting.

[1]Department of Genetics, Blavatnik Institute, Harvard Medical School, Boston, MA 02115, USA. [2]Department of Cancer Biology, Dana-Farber Cancer Institute, Boston, MA 02115, USA. [3]Department of Cell Biology, Blavatnik Institute, Harvard Medical School, Boston, MA 02115, USA. [4]Institut de Recherche en Cancérologie de Montpellier, INSERM, Montpellier, France. [5]Howard Hughes Medical Institute, Boston, MA 02115, USA. [6]School of Life Sciences and Technology, Tongji University, Shanghai, China. ✉e-mail: psaavedra@hms.harvard.edu; perrimon@genetics.med.harvard.edu

In *Drosophila*, several models of tumor-induced organ wasting have been recently described[20–25]. One such model of tumor-induced organ wasting consists of generating tumors in the intestine of adult flies by overexpression of an active form of the transcriptional co-activator *yorkie*/YAP (*yki[S3A]*)[20]. These gut *yki*-tumors secrete an insulin binding protein (IBP), ImpL2[26,27], that reduces insulin signaling in peripheral tissues and induces organ wasting. High levels of tumor-derived ImpL2 associate with impairment of muscle function, reduced ATP content, ovary atrophy, hyperglycemia and reduction of stored triglycerides[20].

Here, using the *yki*-tumor gut model (*Esg>yki[S3A]*)[20], we focused on the flight muscles of the adult fly thorax to identify genes involved in muscle energy metabolism. We identify REPTOR, a transcription factor controlled by the mTOR pathway[28], as a key regulator of glucose metabolism in flies with *yki*-tumors. We show that an increase in REPTOR activity in muscles is sufficient to repress glycolysis and to induce muscle degradation in flies. Finally, we propose that both REPTOR and its mammalian ortholog, CREBRF, work as regulators of metabolic flexibility by repressing glycolysis.

## Results

### *Esg>yki[S3A]* flies show alterations in energy metabolism and myofiber degradation of flight muscles

To study how gut *yki*-tumors affect muscle energy metabolism in adult flies, we measured triglyceride and glucose content in *Esg>yki[S3A]* thoraces over time. The adult thorax is mainly composed of muscle fibers that control wing and leg movement but also has portions of fat body, the fly adipose tissue (Fig. 1a). Triglyceride levels slowly decreased with aging in control thoraces but *Esg>yki[S3A]* thoraces displayed a sharp reduction in triglyceride content after 8 days of tumor induction (Fig. 1b), suggesting increased triglyceride breakdown[20]. Conversely, we observed a progressive increase in tissue glucose content, from 8 to 14 days after tumor induction (Fig. 1c), which may indicate a reduction in glucose usage. Taken together, these results suggest that *Esg>yki[S3A]* flies display changes in lipids and glucose utilization similar to what is observed in cancer patients with muscle metabolic changes[17].

Glucose is the main energy substrate used by flight muscles of *Drosophila*, whereas lipids appear to be unable to fuel continuous flight possibly due to their slow processing by the muscle[29]. Insects that use carbohydrates as an energy source to power flight muscles rely on aerobic glycolysis and oxidative metabolism to convert glucose to pyruvate for fueling the TCA cycle[30–33]. Yet, transcriptomic analysis of thoraces of flies with gut *yki*-tumors show a reduction in the transcriptional signature of both glycolysis and oxidative phosphorylation, while flight muscles exhibit loss of function as evidenced by wing positioning defects and mitochondrial dysfunction[20]. Based on these previous observations, we analyzed by immunostaining the thoracic dorso-longitudinal muscles (DLM), a subgroup of indirect flight muscles, and confirmed the presence of myofiber degradation and compromised sarcomere structure in flies bearing gut *yki*-tumors (Fig. 1d–g, Supplementary Fig. 1a–f). The percentage of *Esg>yki[S3A]* flies showing signs of muscle degradation increased between 14 and 20 days after tumor induction, with most hemi-thoraces exhibiting myofiber degradation by 20 days (Fig. 1h). This correlated with a significant decrease in protein content in *Esg>yki[S3A]* thoraces between 14 and 20 days (Supplementary Fig. 1g). Thus, the onset of myofiber degradation occurs after changes in energy metabolism associated with increased lipolysis and likely reduction in glucose usage.

Mitochondrial dynamics and myofibril structure of *Drosophila* flight muscles are coordinated by a proposed mechanical feedback mechanism[34]. The elongated shape of mitochondria in normal flight muscles changes to a circular morphology due to loss of tension in myofibrils[34]. Furthermore, in *Esg>yki[S3A]* thoraces, aberrant mitochondrial morphology was linked to reduction in ATP content[20].

Immunostaining of degraded myofibers of *Esg>yki[S3A]* thoraces showed mitochondria with circular morphology rather than the elongated shape of control samples (Fig. 1d–g), implying a compromised structure of the flight muscle myofibers. Supporting this observation, *Esg>yki[S3A]* thoraces displayed severe impairment of the t-tubule network[35,36] with a complete lack of staining of *discs large 1* (*dlg1*) and amphiphysin proteins in degraded muscle fibers (Supplementary Fig. 1a–f). Therefore, our results indicate that the muscle fibers in *Esg>yki[S3A]* thoraces undergo severe degradation that associates with alterations in mitochondria morphology and energy metabolism.

In *Esg>yki[S3A]* flies, *ImpL2* produced by the gut *yki*-tumors reduces systemic levels of phosphorylated AKT (pAKT), an indicator of lower insulin signaling[20] (Supplementary Fig. 2a). Given that elevated levels of pAKT induced by Wingless signaling in muscle of *Esg>yki[S3A]* flies associate with a decrease in the percentage of flies with wing position defects[37], we checked whether this rescue was directly related to increasing insulin signaling specifically in muscle. Expression of Myr-istoylated AKT (Myr-AKT) in muscle of *Esg>yki[S3A]* thoraces, while inducing *yki*-tumors in the gut with the *LexA-LexAop* system[38], indeed ameliorated muscle degradation (Fig. 1i), without affecting tumor growth (Supplementary Fig. 2b). Since insulin signaling stimulates glycolysis[39], and the transcriptional profile of glycolysis is reduced in *Esg>yki[S3A]* thoraces[20], we looked for changes in the glycolytic pathway. As expected, the amelioration of the myofiber degradation in *Esg>yki[S3A]* thoraces due to increased *Myr-AKT* in muscle correlated with the transcriptional upregulation of the muscle rate-limiting glycolytic enzymes *Hexokinase-a* (*Hex-a*), *Phosphofructokinase* (*Pfk*) and *Pyruvate kinase* (*Pyk*) (Supplementary Fig. 2c). Further, ATP quantification demonstrated that gut *yki*-tumors reduced thoracic ATP levels, whereas the expression of *Myr-AKT* in muscle of *Esg>yki[S3A]* flies significantly restored ATP content (Supplementary Fig. 2d). Taken together, these results suggest that the protective effect of increasing insulin signaling in muscles of *Esg>yki[S3A]* flies might be due in part to promoting glycolysis.

Cancer patients exhibit changes in glucose and fatty acid utilization in muscle[17], suggesting increased fatty acid oxidation in muscle tissue[16]. An increase in lipid oxidation might therefore be deleterious to muscle tissue in *Esg>yki[S3A]* flies. Indeed, knockdown of *CPT1*, the fatty acid importer to mitochondria, in muscle of *Esg>yki[S3A]* flies led to an amelioration of myofiber degradation (Fig. 1j), without affecting tumor growth (Supplementary Fig. 2e). Taken together, our results indicate that the myofiber degradation observed in *Esg>yki[S3A]* flies occurs secondary to changes in energy metabolism. This likely includes reduced glucose usage and possible alterations in fatty acid import to mitochondria.

### *REPTOR* is upregulated in muscles of flies with *yki*-tumors and modulates glucose metabolism

To identify genes responsible for modulating muscle metabolism in the context of gut *yki*-tumors, we performed a time-course RNA-seq experiment in thoraces spanning a tumor induction phase and a tumor "shutdown" phase by taking advantage of a GAL80 temperature-sensitive transgene (*tub-GAL80[TS]*) (Supplementary Fig. 3a). We analyzed gene expression at day 14, a time point near the onset of myofiber degradation in which glucose content is higher in *Esg>yki[S3A]* thoraces. During the tumor "shutdown" phase, which was initiated at day 14 of tumor induction and lasted until day 38 (Supplementary Fig. 3a), triglycerides and glucose partially returned to control levels, while the reduction in protein content in thoraces was halted, indicating that the systemic effects of *yki*-tumors were ameliorated (Supplementary Fig. 3b–d). Also, the expression of both *yki[S3A]* and *ImpL2* in the gut was restored to control levels (Supplementary Fig. 3e, f), and only 50% of *Esg>yki[S3A]* thoraces showed myofiber degradation, which indicated an amelioration of the myofiber degradation (Supplementary Fig. 3g). Our analysis revealed 241 genes downregulated and 269

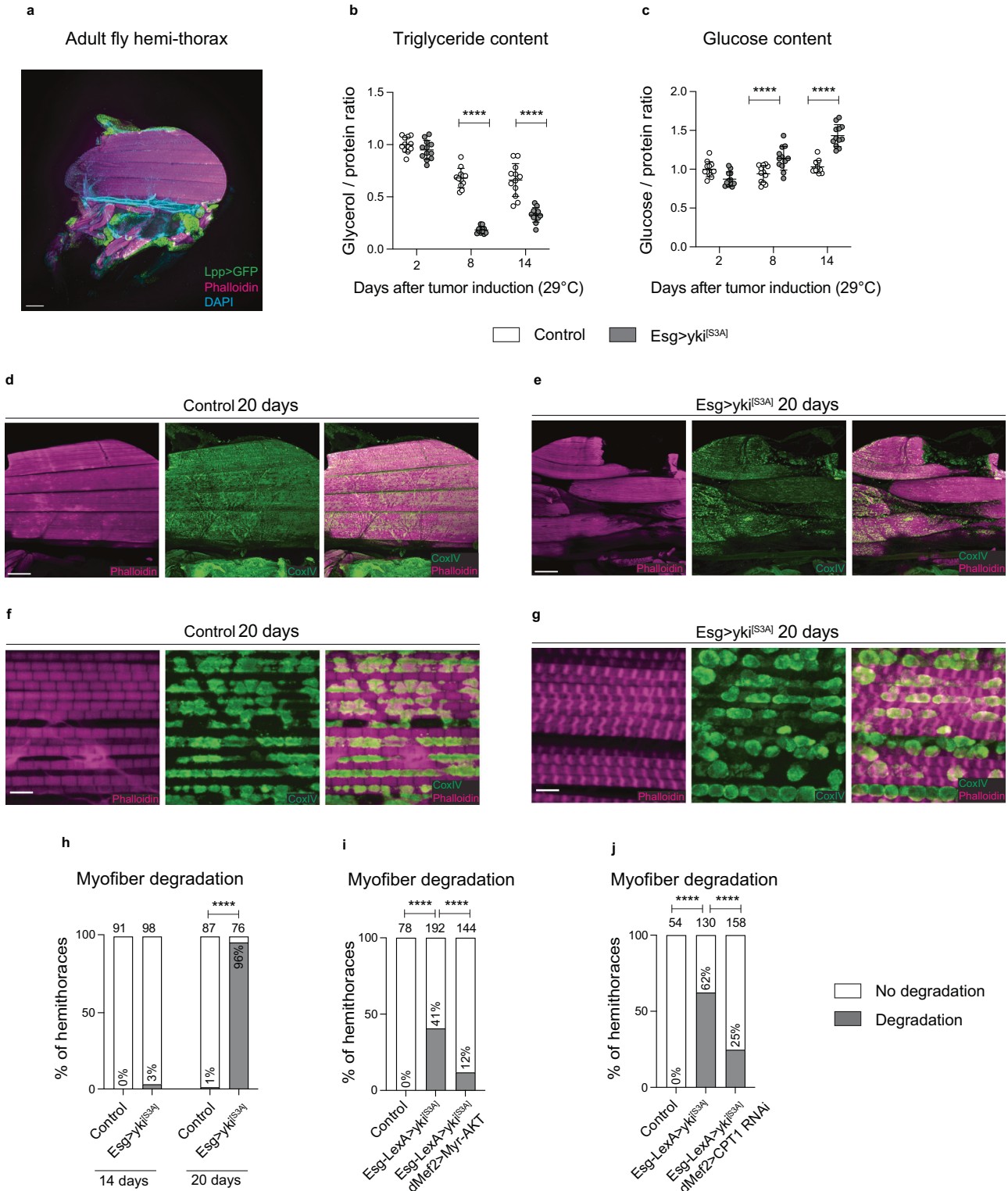

**Fig. 1 | *Esg>yki[S3A]* gut tumors induce alterations in energy metabolism and myofiber degradation. a** Immunostaining of a *Lpp-GAL4* hemi-thorax (wildtype) expressing *UAS-TransTimer*[77] (*Lpp>Transtimer, GFP*) for 4 days to label fat body cells (green) actively expressing *Lpp*. Actin myofibers were stained with phalloidin (magenta). Trachea is observed in the DAPI channel. **b, c** Triglyceride (**b**) and glucose (**c**) content in thoraces (Triglycerides: day 8 and day 14 $p < 0.0001$****. Glucose: day 8 and day 14 $p < 0.0001$****). $N = 12$ biologically independent samples, $N = 8$ thoraces per sample. Triglycerides and glucose were calculated from the same thoraces. Results were reproduced in three independent experiments (**b, c**). **d**–**g** Immunostaining of the flight muscles of hemi-thoraces at different

magnifications (20× **d, e**; 60× **f, g**). Myofibrils were labelled with phalloidin (magenta) and mitochondria with CoxIV (green). **h**–**j** Percentage of hemi-thoraces showing myofiber degradation as shown in (**e, g**) ($p < 0.0001$****). Samples were analyzed at 14 or 20 days (**h**) or 19 days (**i, j**) after tumor induction. Total number of hemi-thoraces scored per genotype is shown. Data shows mean with ±SD (**b, c**). Values were normalized to the mean of control samples of 2 days after tumor induction (**b, c**). Statistical analysis was done using two-way ANOVA with Sidak correction test for multiple comparisons (**b, c**), or two-tailed Fisher's exact test with a confidence interval of 95% for pairwise comparisons between two groups (**h–j**). Scale bar is 100 μm in (**a, d, e**) and 5 μm in (**f, g**). Source data are provided as a Source Data file.

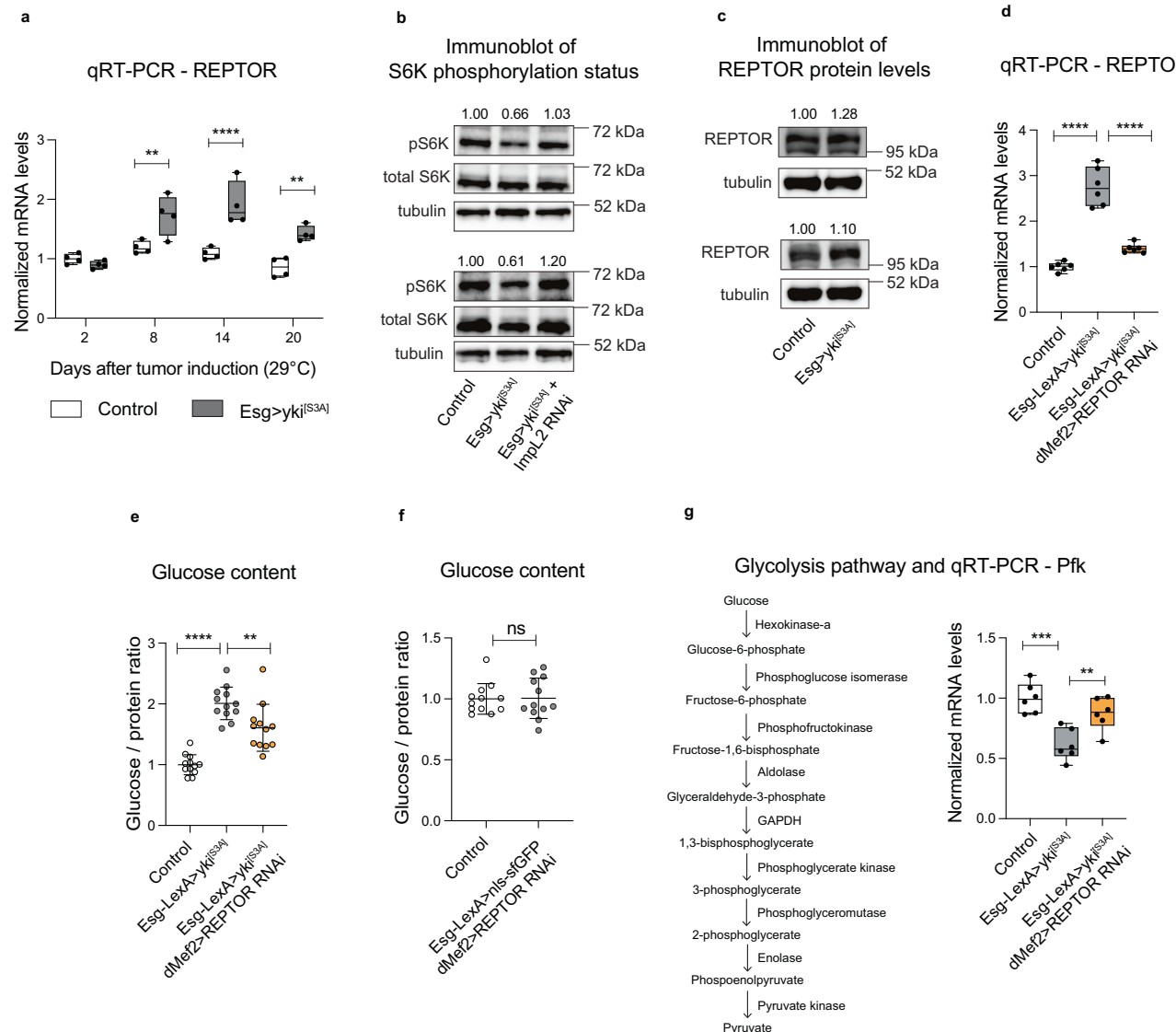

**Fig. 2 | REPTOR regulates glucose metabolism in *Esg>yki[S3A]* thoraces. a** mRNA levels of *REPTOR* in thoraces (8 days $p = 0.004$**, 14 days $p < 0.0001$****, 20 days $p = 0.0025$**). $N = 4$ biologically independent samples, $N = 6–10$ thoraces per sample. **b, c** Protein levels of phosphorylated S6K (pS6K) and total S6K (**b**), or REPTOR (**c**) measured from thoraces after 14 days of tumor induction. $N = 2$ biologically independent samples (**b, c**). Results were reproduced in two (**b**) or three (**c**) independent experiments. Values indicate densitometry of bands normalized to control. **d** mRNA levels of *REPTOR* in thoraces (Ctrl vs *yki[S3A]* $p < 0.0001$****, *yki[S3A]* vs *yki[S3A]* REPTOR RNAi $p < 0.0001$****). $N = 6$ biologically independent samples, $N = 6–10$ thoraces per sample. **e, f** Glucose content in thoraces upon *REPTOR* knockdown in muscle (**e**: Ctrl vs *yki[S3A]* $p < 0.0001$****, *yki[S3A]* vs *yki[S3A]* REPTOR RNAi $p = 0.0033$**. **f**: $p = 0.93$ ns). $N = 12$ biologically independent samples, $N = 8$ thoraces

per sample. Results were reproduced in three independent experiments. **g** Scheme of the glycolytic pathway and mRNA levels of Phosphofructokinase (*pfk*) in thoraces (Ctrl vs *yki[S3A]* $p = 0.0003$***, *yki[S3A]* vs *yki[S3A]* REPTOR RNAi $p = 0.0076$**). $N = 6$ biologically independent samples, $N = 6–10$ thoraces per sample. Samples were analyzed after 12 days of gene induction (**d–g**). Data shows mean with ±SD (**e, f**) or boxplots (median and quartiles) with whiskers (minimum to maximum) (**a, d, g**). Values were normalized to the mean of control samples of 2 days after tumor induction (**a**) or mean of control samples (**d–g**). Statistical analysis was done using two-way ANOVA (**a**), one-way ANOVA with Sidak correction test for multiple comparisons (**d, e, g**), or two-tailed *t*-test with Welch's correction (**f**). Source data are provided as a Source Data file.

genes upregulated during the tumor induction phase, whose expression was then reversed during the "shutdown" phase (Supplementary Data 1). GO-term enrichment and KEGG pathway analysis showed decreased expression of groups of genes related to mitochondria, glycolysis, generation of precursor metabolites and energy, and carbohydrate metabolism during the tumor induction phase (Supplementary Data 1). Some of the increased groups included stress response genes, signal transduction and DNA-transcription factor activity (Supplementary Data 1). Within this last category, we identified 23 transcription factors upregulated during the tumor induction phase that could be candidate regulators of muscle metabolism (Supplementary Fig. 3h).

We focused on *REPTOR*, since it is a transcription factor regulated by TOR signaling, which was previously linked to the regulation of metabolism in *Drosophila*[28]. *REPTOR* mRNA was increased in *Esg>yki[S3A]* thoraces at several time points after tumor induction and was restored to normal levels during the tumor "shutdown" phase (Fig. 2a, Supplementary Fig. 3i).

REPTOR activity was previously shown to be modulated through phosphorylation by the mTOR pathway, leading to its cytoplasmic retention[28]. In conditions of low mTOR activity, REPTOR becomes active and translocates to the nucleus[28]. Given that AKT positively regulates mTOR activity[40–44], and that *ImpL2* produced from gut *yki*-tumors reduces systemic insulin signaling[20], we expected the activity

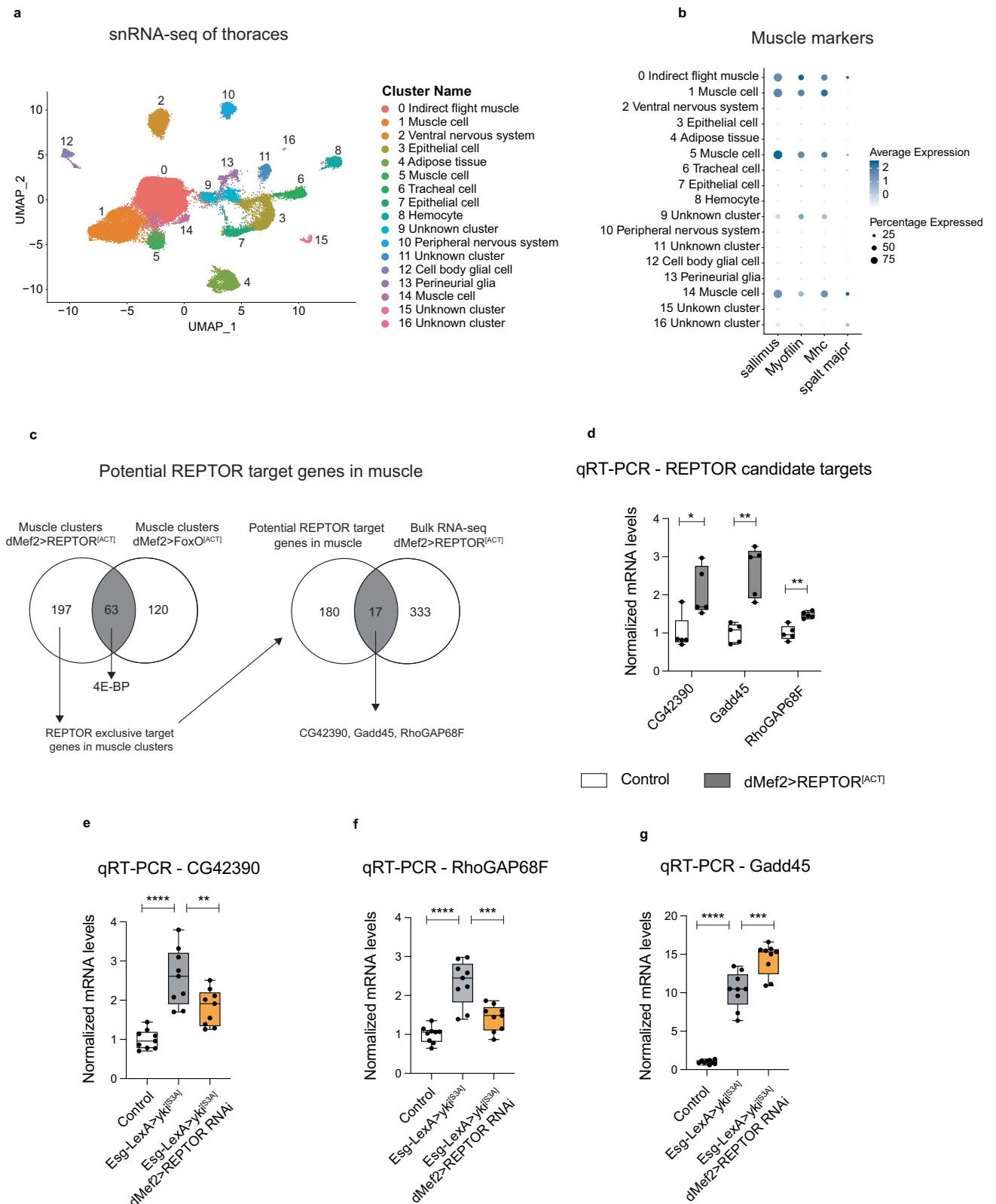

of the mTOR pathway in *Esg>yki[S3A]* thoraces to be reduced. Supporting this hypothesis, we detected in *Esg>yki[S3A]* thoraces a decrease in the levels of phosphorylation of S6K (pS6K), a direct target of mTOR, whereas knockdown of *ImpL2* in *yki*-tumors restored the levels of pS6K (Fig. 2b). Taken together, these observations support the model that mTOR activity is decreased and that REPTOR activity is increased in muscles of *Esg>yki[S3A]* thoraces. Interestingly, we only observed a

marginal increase in protein levels of REPTOR in *yki*-thoraces, suggesting that post-translational modifications of REPTOR may be critical for its activity in the context of *yki*-gut tumors (Fig. 2c).

To test the role of REPTOR in regulating triglyceride and glucose content in thoraces, we decreased the expression of *REPTOR*, and therefore its activity, in muscle using *dMef-GAL4 REPTOR-RNAi*, while inducing *yki*-tumors in the gut with the *LexA-LexAop* system.

**Fig. 3 | Identification of candidate target genes of REPTOR in muscle. a** UMAP plot showing the integration of the datasets of control, *dMef2>REPTOR*[ACT] and *dMef2FoxO*[ACT] thoraces. **b** Dot plot representing the scaled expression of three markers of muscle tissue, *sallimus, Myofilin* and *Myosin heavy chain (Mhc)*. *Spalt major* is used as marker of indirect flight muscles[102]. **c** Schematics of the strategy to identify specific target genes of REPTOR in muscle. All four muscle clusters were included in the analysis. Seventeen genes were identified from the intersection between snRNA-seq muscle clusters and the bulk RNA-seq of *dMef2 > REPTOR*[ACT]. Three candidate targets of REPTOR were shortlisted based on being upregulated in more than one muscle cluster and having REPTOR/REPTOR-BP DNA binding sites. **d–g** mRNA levels in thoraces of the candidate targets identified upon muscle-specific induction of *REPTOR*[ACT] (*CG42390* $p = 0.017*$, *Gadd45* $p = 0.0035**$, *RhoGAP68F* $p = 0.0027**$) (**d**), or knockdown of *REPTOR* in muscle of *Esg>yki*[S3A] flies for 12 days (*CG42390*: Ctrl vs *yki*[S3A] $p < 0.0001****$, *yki*[S3A] vs *yki*[S3A] *REPTOR RNAi* $p = 0.0084**$. *RhoGAP68F*: Ctrl vs *yki*[S3A] $p < 0.0001****$, *yki*[S3A] vs *yki*[S3A] *REPTOR RNAi* $p = 0.0002***$. *Gadd45*: Ctrl vs *yki*[S3A] $p < 0.0001****$, *yki*[S3A] vs *yki*[S3A] *REPTOR RNAi* $p = 0.0002***$) (**e–g**). $N = 5–9$ biologically independent samples, $N = 6–10$ thoraces per sample. Data shows boxplots (median and quartiles) with whiskers (minimum to maximum) (**d–g**). Values were normalized to the mean of control samples (**d–g**). Statistical analysis was done by using two-tailed *t*-test with Welch's correction (**d**), or one-way ANOVA with Sidak correction test for multiple comparisons (**e–g**). Source data are provided as a Source Data file.

Knockdown of *REPTOR* in muscles of *Esg>yki*[S3A] thoraces reduced the elevation of its transcript (Fig. 2d), without affecting tumor growth (Supplementary Fig. 4a). Importantly, we could not retrieve flies after 12 days of tumor induction that maintained a strong suppression of *REPTOR* expression in muscles, demonstrating the critical role of REPTOR for survival of flies with gut *yki*-tumors. Interestingly, *REPTOR* knockdown in muscles did not affect the reduction of triglyceride content in *yki*-thoraces (Supplementary Fig. 4a), indicating that, at least in the context of gut *yki*-tumors, the increased lipolysis induced by tumor-secreted factors[20,45,46] was not dependent on increased *REPTOR* expression in muscle.

Transcriptomic analysis of *Esg>yki*[S3A] thoraces showed a severe reduction in energy metabolism gene signatures, in particular glycolysis[20] (Supplementary Data 1). Importantly, the increase in glucose content observed in *Esg>yki*[S3A] thoraces was significantly blunted after 12 days of tumor induction when *REPTOR* was knocked-down in muscles (Fig. 2e). Further, suppression of *REPTOR* expression in wildtype muscles did not affect glucose levels in thoraces (Fig. 2f). Associated with these results, reducing *REPTOR* expression in *Esg>yki*[S3A] muscles fully restored the expression of *pfk* to control levels (Fig. 2g), while inducing a moderate rescue of *pyk* expression, as well as a slight increase in *hex-a* mRNA levels (Supplementary Fig. 4c). Taken together, our results suggest the involvement of REPTOR in modulating glucose metabolism in muscle of *Esg>yki*[S3A] flies.

### REPTOR is more active in muscles of *Esg>yki*[S3A] flies and modulates the expression of two potential direct target genes *CG42390* and *RhoGAP68F*

To demonstrate that REPTOR was indeed more active in muscles of *Esg>yki*[S3A] thoraces, we checked for changes in expression of potential target genes of REPTOR. To discover specific targets of REPTOR in adult fly muscle, we ectopically expressed the constitutively active allele of REPTOR[28], *REPTOR*[ACT], in muscle and performed single nucleus RNA-seq (snRNA-seq) in dissected thoraces. Our analysis resolved a total of seventeen clusters (Fig. 3a), which were identified using annotations from the Fly Cell Atlas[47]. Of the seventeen clusters, four clusters corresponded to muscle cells, from which cluster 0 was identified as indirect flight muscle (Fig. 3b). We therefore focused our analysis exclusively on these four muscle clusters. Given that FoxO is negatively regulated by insulin signaling[48], and shares around 40% of target genes with REPTOR[28,49], we compared the transcriptional profile of the muscle clusters expressing *REPTOR*[ACT] with muscle clusters expressing a constitutively active allele of *FoxO* (*FoxO*[ACT]). This approach allowed us to pinpoint the transcriptional signature of each transcription factor and exclude common target genes. Consistent with previous results[28], *REPTOR* and *FoxO* expression led to an enrichment of *4E-BP* in muscle clusters (Supplementary Fig. 5a–c), a known target gene of both transcription factors.

Further analysis of gene expression changes in all four muscle clusters revealed 197 REPTOR-specific genes, 120 FoxO-specific genes and 63 common target genes (Fig. 3c, Supplementary Data 2). Next, we looked for REPTOR specific hits that could be detected by qRT-PCR in homogenized thoraces, and shortlisted seventeen genes whose

expression was similarly changed in bulk RNA-seq of *dMef2 > REPTOR*[ACT] thoraces. (Fig. 3c, Supplementary Data 1). From these, we focused on studying three genes that were upregulated in more than one muscle cluster and had DNA binding sites for REPTOR/REPTOR-BP (Supplementary Data 2). *CG42390, RhoGAP68F* and *Gadd45* were confirmed to be upregulated in wildtype thoraces over-expressing *REPTOR*[ACT] in muscle (Fig. 3d), but not in thoraces over-expressing *FoxO*[ACT] (Supplementary Fig. 5d). These genes were also increased in *Esg>yki*[S3A] thoraces, but only *CG42390* and *RhoGAP68F* had their expression reversed when *REPTOR* was knocked-down in muscle (Fig. 3e–g), thus supporting our hypothesis that these genes may act as targets of REPTOR in muscle of the *Esg>yki*[S3A] flies.

Given the role of ImpL2 produced by gut *yki*-tumors in modulating the mTOR pathway in thoraces, we tested whether increased insulin signaling in muscle of *Esg>yki*[S3A] flies would reduce the expression of these potential target genes by increasing mTOR activity and repressing REPTOR. To increase insulin signaling in muscle, we overexpressed *Myr-AKT* with *dMef2*-GAL4, which led, as expected, to an increase in the pS6K/S6K ratio in *Esg>yki*[S3A] thoraces (Supplementary Fig. 5e). In this context, the expression of the *CG42390*, but not of *RhoGAP68F*, was partially restored to control levels (Supplementary Fig. 5f, g), consistent with reduced REPTOR activity.

### Activating REPTOR in wild type muscle phenocopies the changes in glucose content and myofiber structure observed in *Esg>yki*[S3A] flies

Gut *yki*-tumors produce multiple factors that can induce complex systemic metabolic changes[20,45,46], making it difficult to isolate the effects of REPTOR in muscle metabolism. To study the function of REPTOR in wildtype muscles in the absence of gut *yki*-tumors, we tested whether elevating the expression of a wildtype *REPTOR*[WT] allele with *dMef2-GAL4* would induce the same phenotypes observed in *Esg>yki*[S3A] flies. Forced expression of *REPTOR*[WT] increased its protein levels in muscle (Fig. 4a). However, there was no detectable increase in glucose content (Fig. 4b) and no induction of myofiber degradation (Fig. 4c), even when *REPTOR*[WT] was expressed for 20 days (Supplementary Fig. 6a, b).

Since REPTOR is post-translationally regulated by mTOR activity[28], we tested whether the simultaneous expression of *REPTOR* with *PRAS40* or *Tsc1/Tsc2*, two inhibitors of mTOR signaling[43,44,50], could activate REPTOR and induce the changes in energy metabolism and myofiber degradation observed in flies bearing gut *yki*-tumors. Even though the co-expression of *REPTOR*[WT] with *PRAS40* or *Tsc1/Tsc2* led to only a moderate increase in REPTOR protein levels, as compared with *REPTOR*[WT] single expression (Fig. 4a), it induced a significant increase in glucose content (Fig. 4b). Moreover, 70% of thoraces exhibited myofiber degradation (Fig. 4c), with degraded myofibers displaying round-shape mitochondria like those observed in *Esg>yki*[S3A] thoraces (Fig. 4d–g). Importantly, glucose content was unchanged and myofiber degradation absent when *Tsc1/Tsc2* or *PRAS40* was expressed in the absence of *REPTOR*[WT] co-expression (Supplementary Fig. 6c, d). Taken together, these results indicate that sustained activity of REPTOR in muscle tissue is sufficient to collapse

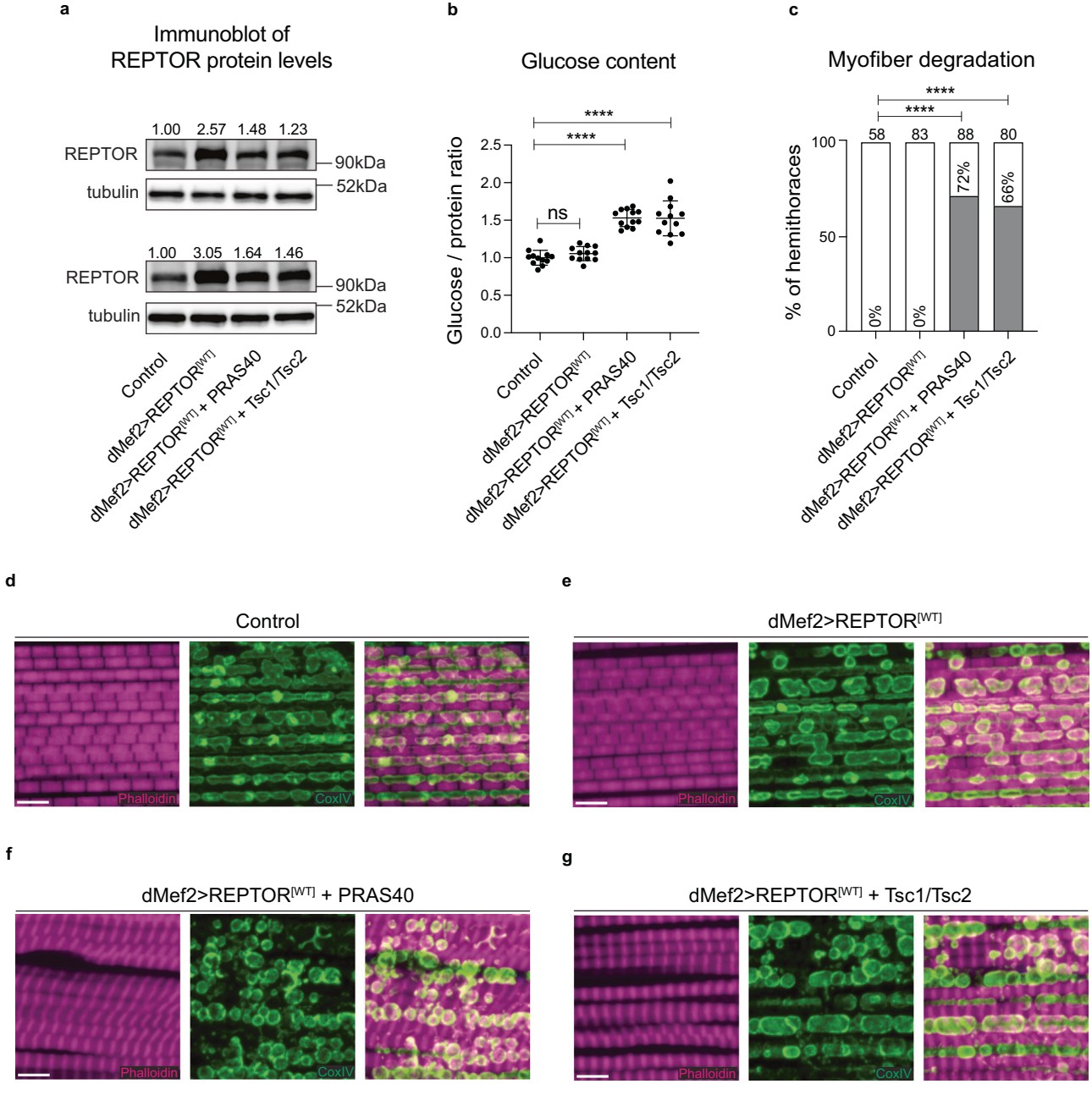

**Fig. 4 | Activation of REPTOR in wildtype muscle increases glucose content and promotes myofiber degradation. a** Protein levels of REPTOR measured from thoraces. $N = 2$ biologically independent samples. Results were reproduced in three independent experiments. Numbers indicate densitometry of bands normalized to control. **b** Glucose content in thoraces (Ctrl vs *REPTOR[WT]* $p = 0.74$ ns, Ctrl vs *REPTOR[WT] + PRAS40* $p < 0.0001$****, Ctrl vs *REPTOR[WT] + Tsc1/Tsc2* $p < 0.0001$****). $N = 12$ biologically independent samples, $N = 8$ thoraces per sample. Results were reproduced in three independent experiments. **c** Percentage of thoraces showing myofiber degradation ($p < 0.0001$****). Total number of hemi-thoraces scored for each genotype is shown. **d–g** Immunostaining of the flight muscles of control (**d**), *REPTOR[WT]* (**e**), or *REPTOR[WT]* co-expressed with *PRAS40* (**f**) or *Tsc1/Tsc2* (**g**). Myofibrils were labelled with phalloidin (magenta) and mitochondria with CoxIV (green). Gene expression was induced for 8 days (**b–g**) Data shows mean with ±SD (**b**). Values were normalized to the mean of control samples (**b**). Statistical analysis was done by using one-way ANOVA with Sidak correction test for multiple comparisons (**b**), or two-tailed Fisher's exact test with a confidence interval of 95% for pairwise comparisons between two groups (**c**). Scale bar is 5 μm in (**d–g**). Source data are provided as a Source Data file.

muscle fibers and induce alterations in glucose metabolism similar to those observed in flies with gut *yki*-tumors.

### Sustained activation of REPTOR in muscle affects energy metabolism by repressing glycolysis

Our results in muscle of both *Esg>yki[S3A]* flies as well as wildtype flies suggest that permanent activation of REPTOR may repress glycolysis, leading over time to myofiber degradation. We therefore used the overexpression of the *REPTOR[ACT]* allele in muscle to assess the

metabolic role of activated REPTOR. As expected, the effects of *REPTOR[ACT]* were similar to those of *REPTOR[WT]* activation by mTOR suppression (see Fig. 4), and included an increase in thoracic glucose content, as well as severe myofiber degradation and alterations in mitochondrial morphology (Supplementary Fig. 7a, b). Notably, analysis of the glycolytic metabolites revealed that *REPTOR[ACT]*-expressing thoraces displayed a significant reduction in the levels of glucose-6-phosphate, fructose-6-phosphate, fructose-1,6-biphosphate and glyceraldehyde 3-phosphate, the first four metabolites of glycolysis

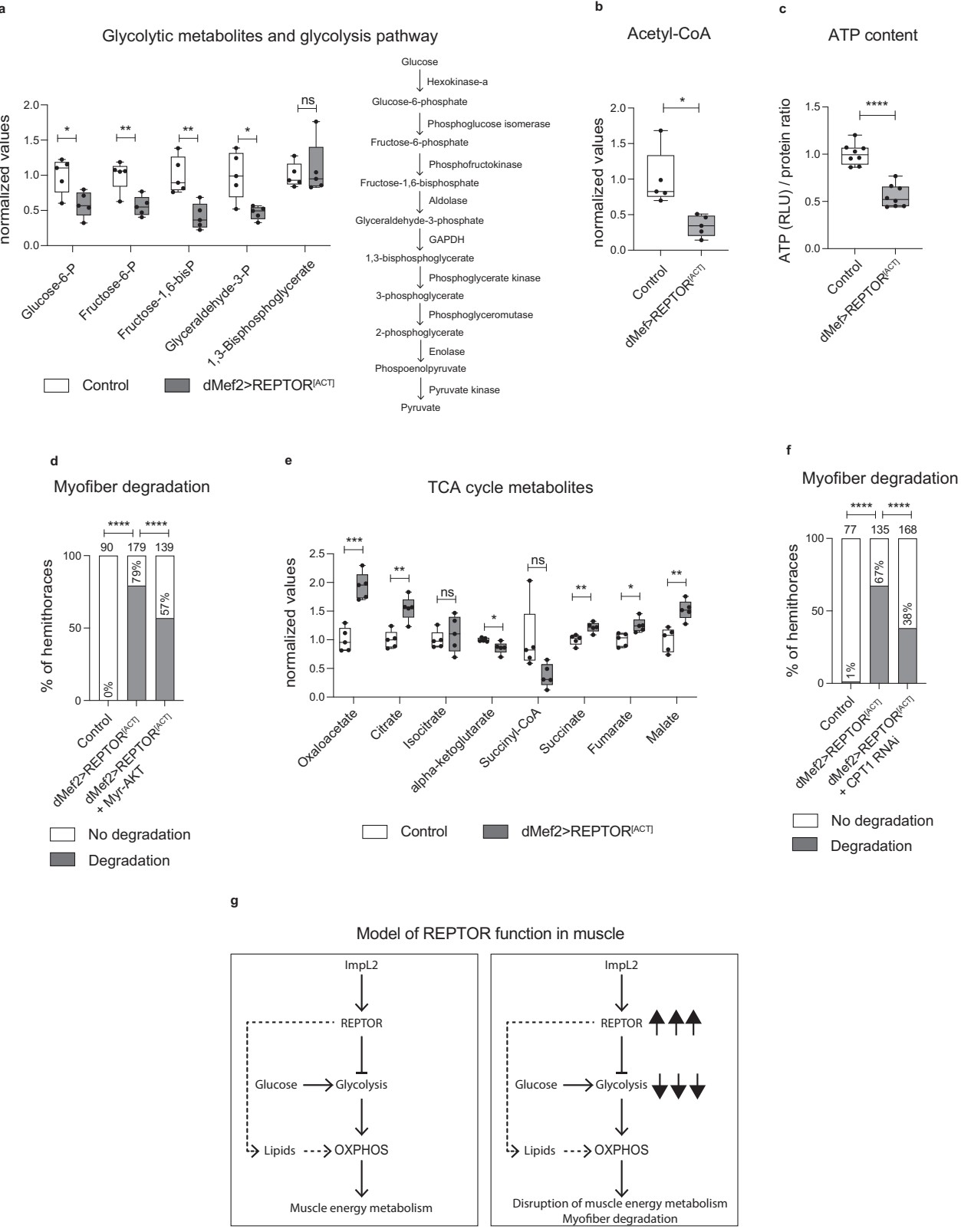

(Fig. 5a). Associated with this reduction, we detected a significant decrease in the expression of the enzymes involved in these glycolytic steps, including *pfk* (Supplementary Fig. 7c). Taken together, these results strongly support that REPTOR is indeed a repressor of glycolysis in muscle tissue.

Next, we investigated the energy metabolism status of the muscle tissue. Consistent with failure of the energy metabolism system, the overexpression of *REPTOR[ACT]* caused a severe reduction in the levels of Acetyl-CoA and ATP content (Fig. 5b, c). Gene set enrichment analysis of RNA-seq from thoraces with sustained expression of *REPTOR[ACT]* in muscles showed a downregulation of genes involved in energy metabolism, in particular glycolysis, but also oxidative phosphorylation and genes encoding mitochondrial proteins (Supplementary Data 1). These results reflect a similar effect to that observed

**Fig. 5 | REPTOR sustained activation in muscle represses glycolysis and collapses the energy metabolism program. a, b** Levels of glycolytic metabolites (Glucose-6-P $p = 0.0196$*, Fructose-6-P $p = 0.0068$**, Fructose 1,6-bisP $p = 0.0042$**, Glyceraldehyde-3-P $p = 0.0224$*, 1,3-Bisphosphoglycerate $p = 0.66$ ns) (**a**), or Acetyl-CoA ($p = 0.0172$*) (**b**) quantified by LC/MS in thoraces. $N = 5$ biologically independent samples, $N = 45$ thoraces per sample. **c** ATP content in thoraces ($p < 0.0001$****). $N = 8$ biologically independent samples, $N = 4$ thoraces per sample. Results were reproduced in three independent experiments. **d** Percentage of hemi-thoraces showing myofiber degradation ($p < 0.0001$****). Total number of hemi-thoraces scored for each genotype is shown. **e** Levels of TCA cycle metabolites quantified by LC/MS in thoraces as in (**a, b**) (Oxaloacetate $p = 0.0002$***, Citrate $p = 0.0021$**, Isocitrate $p = 0.55$ ns, alpha-ketoglutarate $p = 0.0334$*, Succinyl-CoA $p = 0.075$ ns, Succinate $p = 0.0044$**, Fumarate $p = 0.0112$*, Malate $p = 0.0027$**).

$N = 5$ biologically independent samples, $N = 45$ thoraces per sample. **f** Percentage of hemi-thoraces showing myofiber degradation ($p < 0.0001$****). Total number of hemi-thoraces scored for each genotype is shown. **g** Model of the function of REPTOR in muscles. ImpL2 promotes *REPTOR* expression. REPTOR activity represses glucose usage in muscle, while likely promoting the use of lipids as an alternative fuel substrate. REPTOR sustained activation in muscle collapses the energy metabolism program. Data shows boxplots (median and quartiles) with whiskers (minimum to maximum) (**a–c, e**). All values were normalized to the mean of control samples (**a–c, e**). Statistical analysis was done using two-tailed *t*-test with Welch's correction (**a–c, e**), or two-tailed Fisher's exact test with a confidence interval of 95% for pairwise comparisons between two groups (**d, f**). Source data are provided as a Source Data file.

both in the transcriptional profile of the *Esg>yki[S3A]* thoraces[20] (Supplementary Data 1), and on the reduction of ATP content (see Supplementary Fig. 2d). Importantly, the co-expression of *Myr-AKT* with *REPTOR[ACT]* slightly improved the muscle degradation (Fig. 5d). Since a major effect of AKT activation is to stimulate glycolysis[39,51], this result suggests that changes in glucose utilization are at least partially responsible for the deleterious effect of prolonged REPTOR activation in muscle. Thus, our observations imply that sustained activation of REPTOR in muscle is pathological and leads to a detrimental metabolic state that associates repression of glycolysis to energy metabolism collapse and muscle degradation.

In mammals, lipids can be used as an alternative substrate to produce energy in settings in which glucose usage is reduced such as nutritional restriction[5]. Nevertheless, it is unclear whether *Drosophila* flight muscles can rely extensively on lipid utilization[29]. Given that *REPTOR* activation in muscle promoted a moderate reduction in triglyceride content (Supplementary Fig. 7d), it is possible that the suppressive effect of REPTOR on glycolysis permits an elevated use of lipids. Contrasting with the pathway enrichment analysis showing a downregulation of genes involved in oxidative phosphorylation (Supplementary Data 1), LC/MS analysis of *REPTOR[ACT]*-expressing thoraces revealed a significant elevation in several of the TCA cycle metabolites, which may suggest a stimulation of the TCA cycle (Fig. 5e). Importantly, the myofiber degradation induced by sustained REPTOR activation in muscle was ameliorated when the *CPT1* fatty acid transporter was simultaneously knocked-down in muscle tissue (Fig. 5f). These observations suggest that the collapse of the muscle fibers might be due, in part, to changes in fatty acid import to mitochondria. Thus, our results are suggestive of *REPTOR* working as regulator of energy metabolism by strongly suppressing glycolysis while potentially promoting the use of lipids as another fuel substrate (Fig. 5g).

### *FoxO[ACT]* overexpression in muscle does not phenocopy the overexpression of *REPTOR[ACT]*

The systemic metabolic phenotypes in peripheral tissues of *Esg>yki[S3A]* flies are mediated by ImpL2 produced by the gut *yki*-tumors and associate with systemic reduction of insulin signaling[20]. Consistent with previous results[20], *ImpL2* suppression in gut *yki*-tumors improved the reduction in triglyceride content[20], and also ameliorated the glucose content increase and the myofiber degradation observed in *Esg>yki[S3A]* thoraces (Supplementary Fig. 8a–c). Thus, these results indicate that tumor-derived *ImpL2* plays a major role in the two phenotypes associated with sustained REPTOR activation in muscle.

FoxO activity as a transcription factor is repressed by insulin signaling[48], and shares multiple target genes with REPTOR[28,49]. It has also been shown to increase proteolysis in flight muscles[52,53]. We therefore investigated whether FoxO could have a redundant role with REPTOR in driving the muscle metabolic phenotypes observed in *Esg>yki[S3A]* flies. First, we examined whether FoxO was more active in the muscle of flies bearing gut *yki*-tumors. To do so, we used the

snRNA-seq analysis (see Fig. 3) to identify potential FoxO target genes that were not regulated by REPTOR, and which could be used as proxy for FoxO activity in muscle (Supplementary Fig. 8d). Three such targets, *CG10383, l(1)G0469* and *CG1673*, were increased in wildtype thoraces when *FoxO[ACT]* was overexpressed in muscle, but not with *REPTOR[ACT]* overexpression (Supplementary Fig. 8e, f). Importantly, the mRNA levels of these genes was similarly increased in *Esg>yki[S3A]* thoraces, whereas elevating insulin signaling in the muscle of *Esg>yki[S3A]* flies significantly restored the expression of *CG10383* and *l(1)G0469* (Supplementary Fig. 8g). Thus, these observations suggest that FoxO activity may be elevated in muscle of *Esg>yki[S3A]* flies.

Next, we tested whether overexpression of *FoxO[ACT]* in wildtype muscle could induce the same phenotypes than *REPTOR[ACT]*. However, this genetic manipulation did not have a substantial effect on myofiber degradation or in glucose content, even after 20 days of induction (Supplementary Fig. 8h, i). Altogether, these results indicate that REPTOR sustained activity in muscle, but not FoxO, is the main driver of the muscle metabolic phenotypes in *Esg>yki[S3A]* flies.

### Systemic reduction of insulin signaling upregulates *REPTOR* in wildtype flies

Insulin signaling orchestrates pathways to regulate glucose metabolism in muscle tissue in response to changes in nutritional status[54]. Given the opposite roles of insulin signaling and REPTOR in regulating glucose metabolism in muscle, we investigated whether *REPTOR* transcription could be modulated by changes in insulin signaling status. In *Esg>yki[S3A]* thoraces, a setting of reduced insulin signaling, *REPTOR* expression was enhanced by tumor-derived ImpL2 (Supplementary Fig. 9a). However, increasing insulin signaling by expression of *Myr-AKT* specifically in the muscle of these flies failed to strongly suppress the upregulation of *REPTOR* (Supplementary Fig. 9b). These observations suggest that additional systemic effects of ImpL2, on multiple tissues and/or on other signaling pathways, are necessary for its effect on *REPTOR* expression. Consistent with this idea, reducing insulin signaling specifically in muscle of wildtype flies, by overexpressing *p60*, a strong inhibitor of insulin signaling[55], was not sufficient to elevate the levels of *REPTOR* in thoraces (Supplementary Fig. 9c). Likewise, this genetic manipulation did not promote myofiber degradation or elevated glucose content in thoraces (Supplementary Fig. 9d, e).

The pathophysiological nature of the *Esg>yki[S3A]* flies prompted us to investigate the link between insulin signaling and REPTOR in wildtype flies instead. *ImpL2* is involved in reducing systemic insulin signaling during nutritional restriction[26,27], a setting in which glucose usage is reduced. We observed that after a short period of starvation, *REPTOR* expression was induced in thoraces (Supplementary Fig. 9f). Interestingly, *pfk* expression in starved flies was increased when *REPTOR* was suppressed in muscles, indicating that REPTOR suppresses *pfk* in a context of decreased glucose usage (Supplementary Fig. 9g). Supporting the association between nutritional restriction and *REPTOR* expression, systemic reduction of insulin signaling in a

tumor-free setting through ectopic expression of *ImpL2* in normal guts, also led to an upregulation of *REPTOR* in thoraces (Supplementary Fig. 9h). Importantly, in wildtype flies, ectopic expression of *ImpL2* in guts was not sufficient to promote myofiber degradation or alter glucose content (Supplementary Fig. 9i, j). Nevertheless, these results indicate that *REPTOR* expression is induced by changes in the nutritional status of an organism that are associated with reduced glucose utilization.

### CREBRF, the mammalian ortholog of REPTOR, drives a shift from glycolysis to oxidative metabolism in mouse myotubes

To assess the evolutionary conservation of REPTOR function in the regulation of muscle metabolism, we examined CREBRF, the mammalian ortholog of REPTOR. The human CREBRF protein shares 16% identity and 25% similarity with REPTOR, although the REPTOR residues phosphorylated by mTOR (S527 and S530) do not appear to be conserved in CREBRF. Nevertheless, *Crebrf* expression has been reported to increase upon mTOR inhibition, and its forced expression protects against starvation in cultured adipocytes, suggesting that its activity may be responsive to nutrient availability[56,57]. Moreover, a *Crebrf* coding polymorphism associated with obesity in the Samoan population has the strongest effect on BMI of any common obesity-risk variant[57].

We first examined the regulation of the endogenous *Crebrf* gene. In mice, *Crebrf* transcript was increased in muscle upon fasting and suppressed upon refeeding, confirming its regulation by nutritional availability (Fig. 6a). *Crebrf* elevation in muscle was also associated with cancer anorexia cachexia syndrome in a mouse genetic model of non-small cell lung cancer (Supplementary Fig. 10a)[58]. In C2C12 mouse myotubes in vitro, *Crebrf* was increased upon the stress of nutrient withdrawal (Fig. 6b), whereas it was decreased by insulin signaling (Fig. 6c).

Next, we examined the effects of forced *Crebrf* expression on energy metabolism in myotubes. We transduced C2C12 myotubes with adenovirus encoding *Crebrf* and measured the relative contributions of mitochondrial respiration and glycolysis to overall cellular ATP production. In control cells, glycolysis and mitochondrial respiration each contributed approximately half of the cellular ATP production. In contrast, myotubes expressing *Crebrf* showed a substantial shift away from glycolysis: their absolute rate of glycolytic ATP production decreased by 60%, alongside a concomitant 40% increase in the absolute rate of mitochondrial ATP production (Fig. 6d). This effect was independent of the energy substrates available to myotubes, as it was observed not only in the presence of glucose, pyruvate, and glutamine but also in media lacking all three of these components (Supplementary Fig. 10b). As expected, lactate production was halved in myotubes expressing *Crebrf*, consistent with their reduced glycolytic ATP production (Fig. 6e). These cells also exhibited reduced glucose uptake and increased glycogen content (Fig. 6f, g), further suggestive of reduced glycolytic flux. In an independent model of myotubes derived from primary mouse myoblasts, *Crebrf* caused similar metabolic changes: a 60% reduction in glycolytic ATP production alongside reduced lactate production, reduced glucose uptake, and mildly increased cellular glycogen (Supplementary Fig. 10c–f).

Given the role of CREBRF in transcriptional regulation, we next used RNA sequencing to assess the gene expression changes associated with the altered energy metabolism of C2C12 myotubes expressing *Crebrf*. Unbiased gene set enrichment analysis revealed oxidative phosphorylation metabolism as the top upregulated term, whereas the top downregulated terms were interferon response, TNFα signaling, and hypoxia (Supplementary Fig. 10g). Moreover, the glycolysis gene set itself was significantly decreased (Supplementary Fig. 10h). The concerted upregulation of mRNAs encoding many mitochondrial proteins suggested a change in activity of an upstream regulator. Indeed, the protein levels of PGC1α, a transcriptional co-activator sufficient to drive mitochondrial biogenesis, were increased

in cells expressing *Crebrf*, as were mRNA levels of known PGC1α target genes (Supplementary Fig. 10i, j, Supplementary Data 1). Together, these results suggest that *Crebrf* drives gene expression changes that favor mitochondrial respiration over glycolysis in myotubes.

## Discussion

Dynamic regulation of glycolysis and oxidative phosphorylation in energy metabolism is critical for adaptation to changes in nutritional availability, and disruption of this flexibility is often associated with metabolic diseases, such as obesity and insulin resistance[7–9], or in cancer patients[17]. Using a model of gut *yki*-tumors in *Drosophila* that shows signs of impaired energy metabolism and muscle degeneration[20], we identify REPTOR as a key player in driving the muscle metabolic phenotypes described in these flies. We provide evidence that in the pathophysiological context of tumors, REPTOR is more active in muscle and disrupts glucose metabolism. We uncouple the effect of REPTOR from the context of gut *yki*-tumors and demonstrate that in wildtype flies, sustained activation of REPTOR in muscle represses glycolysis and induces muscle degradation. We uncover that CREBRF, the mammalian ortholog of REPTOR, is also a potent repressor of glycolysis in mouse myotubes. We propose that REPTOR and CREBRF act as regulators of metabolic adaption to nutrient availability in muscle by repressing glucose utilization, thereby allowing, directly or indirectly, the use of distinct fuel sources such as lipids.

Flight muscles across multiple species of insects use predominantly aerobic glycolysis and oxidative phosphorylation to produce ATP[30–33]. In *Drosophila*, carbohydrates, in particular glucose, are the most efficient energy source to power flight[29]. Permanent activation of REPTOR in muscle, as in flies bearing gut *yki*-tumors or in flies with forced expression of *REPTOR*[ACT], eventually led to the collapse of the energy metabolism program with reduced ATP content in thoraces and muscle degradation. Disruption of glycolysis affects the wing beat or flight ability in *Drosophila*[59,60]. Consistent with these observations, sustained REPTOR activity in muscles induced a sharp reduction in the levels of glycolytic metabolites. Conversely, the expression of *Myr-AKT* in muscle improved the myofiber degradation observed in *Esg>yki*[S3A] flies or associated with *REPTOR*[ACT] expression. Although AKT affects multiple targets[51], one of its major functions is the stimulation of glycolysis[39], suggesting that re-activation of glycolysis can counteract the deleterious effect of REPTOR permanent activation in muscle. Nevertheless, in a pathophysiological context of gut *yki*-tumors, sustained REPTOR activity in *Drosophila* muscles might be adaptive, serving as a systemic response that limits glucose use in muscle to prioritize its availability to other organs that rely on glucose, such as the nervous system[61].

Disruption of insulin signaling is often observed in cancer and impairs glucose metabolism[14]. Our results place REPTOR under the control of ImpL2, a secreted factor from gut *yki*-tumors that reduces systemic insulin signaling. In *Esg>yki*[S3A] flies, ImpL2-derived from tumors is linked to transcriptional upregulation of REPTOR in thoraces, as well as to increased glucose content and muscle degradation, two phenotypes induced by sustained activity of REPTOR in muscle. Nevertheless, in wildtype flies, systemic reduction of insulin signaling or muscle-specific decrease in mTOR pathway activity did not phenocopy the effect of *REPTOR*[ACT] overexpression. These observations reflect the complexity of the *Esg>yki*[S3A] flies, in which multiple molecules are being secreted by the gut *yki*-tumors (i.e., ImpL2, Pvf1, Upd3) that can induce signaling pathways systemically[45,46], and may additionally affect REPTOR activity, besides the mTOR-mediated post-translational regulation.

Systemic reduction of insulin signaling in wildtype flies upregulated *REPTOR* in thoraces, whereas muscle-specific reduction of insulin signaling did not. It is possible that ImpL2 modulates *REPTOR* expression by reducing insulin signaling in other peripheral tissues

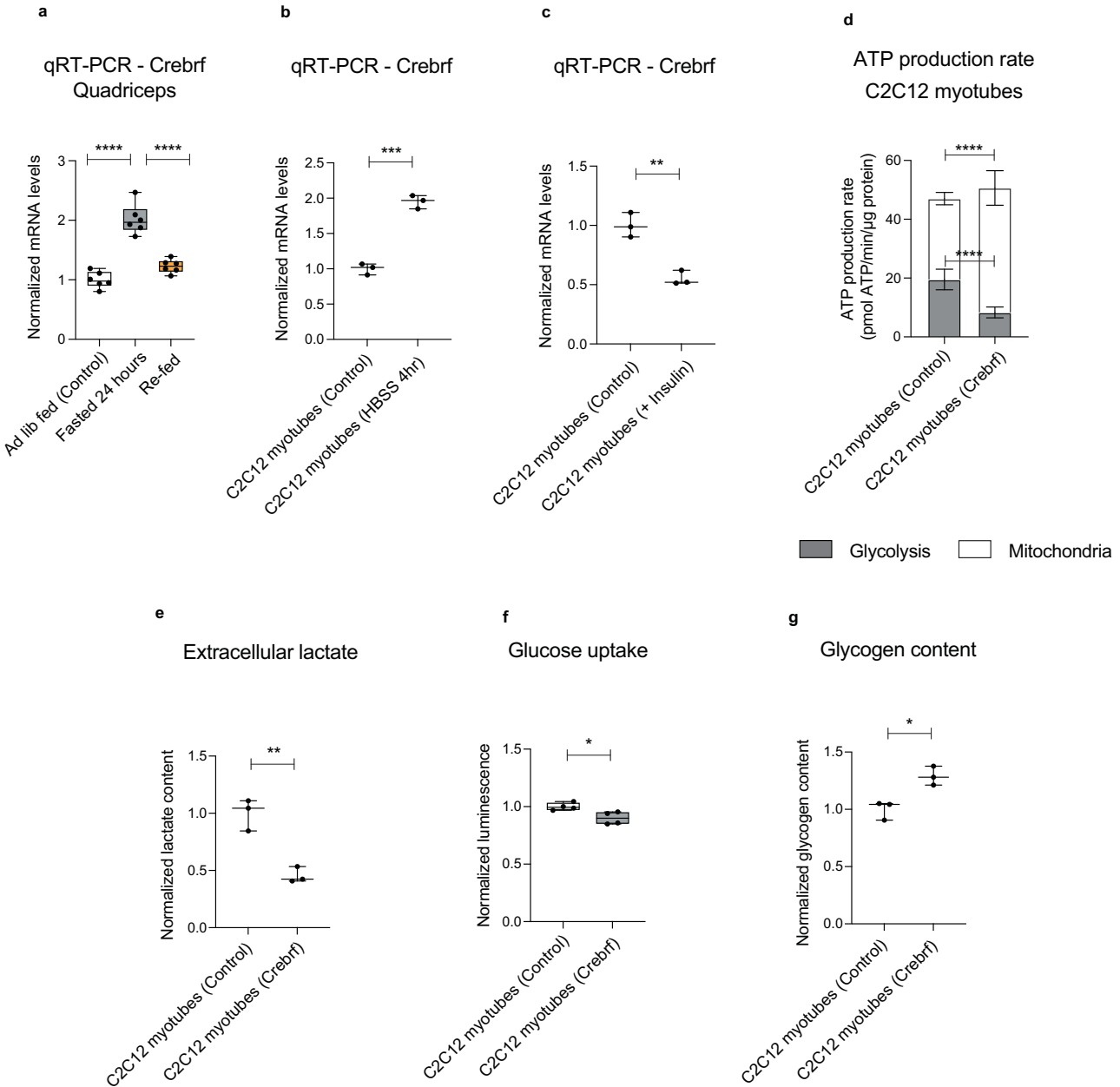

**Fig. 6 | CREBRF represses glycolysis and promotes oxidative metabolism in mammalian myotubes. a–c** *Crebrf* mRNA levels in: mouse quadriceps muscle upon *ad libitum* feeding, 24 h fast or 4 h re-feeding (Ctrl vs Fasted $p < 0.0001$****, Fasted vs Re-fed $p < 0.0001$****) (**a**), C2C12 myotubes incubated in HBSS for 4 h ($p = 0.0002$***) (**b**) or C2C12 myotubes treated with insulin (100 nM) ($p = 0.006$**) (**c**). $N = 3–6$ biologically independent samples. **d** ATP production rate contributed by glycolysis and mitochondrial respiration in C2C12 myotubes upon adenoviral expression of *Crebrf*. (Glycolysis $p < 0.0001$****, Mitochondria $p < 0.0001$****). Values for glycolysis and mitochondrial respiration were calculated from $N = 10$ biologically independent bioanalyzer wells for each condition. **e** Secreted lactate measurement in media conditioned by C2C12 myotubes expressing *Crebrf* ($p = 0.0092$**). **f** Glucose uptake in myotubes expressing *Crebrf* ($p = 0.027$*). **g** Glycogen content in myotubes expressing *Crebrf* ($p = 0.0125$*). $N = 3–4$ biologically independent samples. Results were reproduced in two independent experiments (**b–g**). Data shows mean with ±SD (**d**) or boxplots (median and quartiles) with whiskers (minimum to maximum) (**a–c, e–g**). All values were normalized to the average of control samples (**a–c, e–g**). Statistical analysis was done using one-way ANOVA with Sidak correction test for multiple comparisons (**a**), or two-tailed *t*-test with Welch's correction (**b–g**). Source data are provided as a Source Data file.

associated with metabolic regulation. One such tissue may be the fat body, where lipolysis is regulated by insulin signaling[62,63]. Supporting this notion, lipolysis is increased in flies with gut *yki*-tumors[20] or under nutritional restriction[62], both conditions in which *REPTOR* is transcriptionally induced. Thus, REPTOR regulation by insulin signaling may be an adaptive, and likely temporary, response to changes in nutrient availability, to prevent glucose usage in muscle while promoting a switch to other available energy substrates. However, in disease conditions such as the gut *yki*-tumors, permanent activation of

REPTOR caused by systemic dysregulation of insulin signaling becomes pathological to the muscle.

Human genetics suggested a metabolic role for CREBRF in the control of cellular energy utilization[57]. A coding polymorphism of CREBRF (R457Q) is common among Pacific populations and has been identified as one of the strongest known variants linked to obesity[57]. Our results provide a global view of gene expression programs regulated by CREBRF and demonstrate that increased levels of CREBRF favor mitochondrial oxidative metabolism in vitro. This shift is

achieved in part through repression of glycolysis, resulting in a switch towards mitochondrial respiration that is accompanied by an increase in the oxidative phosphorylation gene expression program. The fact that CREBRF and REPTOR can repress glycolysis in muscle may enable the use of alternative energy substrates to fuel oxidative phosphorylation, like lipids, allowing an adaptation to settings of unfavorable nutrient availability. Consistent with this, our results in mice show that *Crebrf* levels increase in muscle under nutritional restriction—a setting in which fatty acid oxidation is increased and glucose usage reduced— and return to control levels after re-feeding. REPTOR showed a similar transcriptional upregulation in starved flies.

Mouse modes of cachexia demonstrated that excessive lipolysis or fatty acid oxidation can contribute to alterations of muscle metabolism and muscle atrophy[64–67]. Glycolytic myofibers are more severely affected in cachexia, contrasting with oxidative muscle fibers that are more resistant to wasting[58]. In this context, the activity of glycolytic enzymes is reduced in glycolytic muscle fibers, as is glucose oxidation[58]. The increase in *Crebrf* observed in the gastrocnemius, a predominantly glycolytic muscle, of cachectic mice may contribute to this phenomenon. Whether CREBRF ultimately contributes to muscle wasting or protection remains to be determined, and may depend on the type of myofiber involved and the nature of the other mitochondrial fuel sources available to the muscle.

Disruption of the adaptive capacity of muscle tissue to switch between lipids and glucose oxidation has been proposed to cause a metabolic gridlock in mitochondria that disrupts energy metabolism[68]. In insects, locusts can use lipids to sustain long-term flight[69–71], but it is not clear what role lipids play in powering muscles of *Drosophila*[29]. Flies maintained on a high-fat diet were shown to have impaired mitochondria respiration and lower ATP content in thoraces[72]. Consistent with these observations, repressing the fatty acid importer *CPT1* in a setting of sustained REPTOR activity reduced the percentage of flies showing myofiber degradation. These results suggest that alterations in the flux of fatty acid import to mitochondria of flight muscles may be linked to their collapse. In mammals, changes in fatty acid oxidation in heart muscle have been implicated in the cardiac function in heart failure and ischemic heart disease[73]. Our results thus provide molecular insight into the adaption of muscle tissue to changes in nutrient availability.

## Methods

### *Drosophila* stocks

Bloomington *Drosophila* Stock Center (BDSC): *esg.LexA::GAD* (P{ST.lexA::HG}SJH-1, 66632). *UAS-p6O* (P{UAS-Pi3K21B.HA}2, 25899). *UAS-Myr-Akt* (P{UAS-myr-Akt.ΔPH}3, 80935). *tub-GAL80[TS]* (P{tubP-GAL80[ts]}10, 7108. P{tubP-GAL80ts}7, 7018). *UAS-CPT1 RNAi* (P{y[+t7.7] v[+t1.8]=TRiP.HMS00040}attP2, 34066). National Institute of Genetics Stock Center (NIG): *UAS-ImpL2-RNAi* (15009R-3). *UAS-REPTOR-RNAi* (13624R-3). Vienna *Drosophila* Research Center (VDRC): *Act88f::GFP* (PBac{fTRG10028.sfGFP-FT} v318362). Laboratory stocks: *w[1118]*. *tub-GAL80[ts]*. *dmef2-GAL4. tub-GAL80[ts]; lpp-GAL4. esg-GAL4, UAS-GFP, tub-GAL80[ts][74]. UAS-yki[S3A][75]. UAS-s.ImpL2[27]*. UAS-*Tsc1/Tsc2[76]. UAS-Transtimer[77]*. The stock *UAS-Empty[VK33]* (3[rd] chromosome) was a gift from Hugo Bellen laboratory. The stocks *UAS-REPTOR[ACT][28]* and *UAS.PRAS40[78]* were a gift from Aurelio Teleman. The stocks *UAS-HA-FoxO[ACT], UAS-REPTOR[WT], LexAop-nls-sfGFP* and *LexAop-yki[S3A]-sfGFP* were made in this work.

### Crosses

Flies were kept on standard cornmeal fly food supplemented with agar and yeast. Around 20 virgin females were used per cross and each cross was kept at 18 °C under similar conditions of food. Bottles were flipped every 4–6 days. For gut *yki*-tumors, male flies were collected every 24–48 h and then incubated at 29 °C (*GAL4*) or 27 °C (*LexOp*) for induction of *yki[S3A]* expression in the gut. For muscle-specific

expression (*dMef2-GAL4*), adult males were collected every 48–72 h, kept at 18 °C for an additional 4–5 days and then incubated at 29 °C to induce gene expression. Muscle-specific overexpression was done for 4 days unless stated otherwise. Up to 20 males were kept in each vial to avoid overcrowding. Vials at 29 °C were flipped every 2–3 days. For a list of *Drosophila* genotypes used for each figure see Supplementary Data 3.

### Nutritional restriction

Adult males were collected every 48 h, kept for 5 days at 18 °C and then incubated at 29 °C for 24 h to induce gene expression prior to starvation. Flies were then transferred to vials with agarose 0.8% supplemented with 2% sucrose (Fisher Scientific) and maintained at 29 °C for 3 days.

### DNA and plasmids

PCR amplification of DNA from plasmids or fly genomic DNA was done using Phusion polymerase (NEB−M0530). Digestion of plasmids with restriction enzymes was done at 37 °C for 2–3 h. Linearized plasmids and PCR fragments were gel purified using QIAquick columns (QIAGEN − 28115). PCR fragments and plasmid backbones were assembled using Gibson assembly (NEB−E2611). For Gateway cloning, Gateway-compatible expression and entry vectors were recombined using LR Clonase II (Thermo Fisher Scientific − 11791020). Plasmid DNA was extracted from bacterial cultures using a QIAprep Spin Miniprep Kit (QIAGEN − 27104) and Sanger sequenced at the Dana-Farber/Harvard Cancer Center DNA Resource Core or Genewiz.

### Generation of the *LexAop-nls-sfGFP* and *LexAop-yki[S3A]-sfGFP* fly stocks

The nuclear superfolder GFP[79] (*nls-sfGFP*) construct was amplified from a *T2A-sfGFP* plasmid kindly provided by Justin Bosch (unpublished) with the following oligos: forward primer "CTTCAGGCGGCCGCGGCCAACATGCCCAAGAAGAAGCGCAAGGTGGTGTCCAAGGGCGAGGAGC" and reverse primer "CTTCACAAAGATCCTCTAGCTACTTGTACAGCTCATCCATG". The *yki[S3A]* sequence was amplified from genomic DNA of *UAS-yki[S3A]* flies[75] using as forward primer "GTACGAATTCCAACATGTTAACGACGATGTCAGCC" and the reverse primer "ATATGCGGCCGCTCACGTAGAATCGAGAC" that includes a homologous sequence to the V5-tag present in the *yki[S3A]* transgene[75]. The *yki[S3A]* PCR fragment was cloned in frame with *sfGFP* (Justin Bosch−unpublished). The *yki[S3A]-sfGFP* fragment was amplified using the following oligos: forward primer "CCTTTACTTCAGGCGGCCGCGGCCACCATGTTAACGACGATGTCA" and reverse primer "CTTCACAAAGATCCTCTAGCTACTTGTACAGCTCATCCATG". The *nls-sfGFP* and *yki[S3A]-sfGFP* PCR products were then cloned by Gibson assembly in the backbone of the pJFRC19-13XLexAop2-IVS-myr::GFP plasmid (Addgene − 26224) previously digested with XbaI and XhoI to remove the *myr::GFP* sequence. Final plasmids were injected in fly embryos of the stock *yw; nos-Cas9 attP40* for targeted insertion in the second chromosome.

### Generation of *UAS-HA-FoxO[ACT]* line

The complete cDNA of *dFOXO* was obtained from the BDGP collection, clone LD19191 (accession no. AF426831), and subcloned into a gateway entry vector using the following oligo sequences: forward primer "CACCATGATGGACGGCTACGCGCAGGAATG" and reverse primer "CTAGTGCACCCAGGATGGTGGCGAGGTCAC". The Akt phosphorylation sites in *dFOXO* were mutated to T44E, S190E, and S259E using the oligos sequences: forward/reverse primers "CGGGCCAGATCCAACGAGTGGCCATGTCCGCG"/ "CGCGGACATGGCCACTCGTTGGATCTGGCCCG" for T44E; "CGCCGCCGTGCCGCTGAGATGGAGACGTCCCGG"/ "CCGGGACGTCTCCATCTCAGCGGCACGGCGGCG" for S190E; and "CGGCAACGCGCCCTCAGAGAATGCCAGTTCCTGC"/ "GCAGGAACTGGCATTCTCTGAGGGCGCGTTGCCG" for the S259E position, according to the manufacturer's procedure for the multisite mutagenesis kit

(Invitrogen). The resulting construct was cloned into the *pHWUASt* destination vector from the *Drosophila* Gateway Vector collection to generate a N-terminal HA-tagged *FoxO*. Transgenic flies were obtained by injecting the construct into *w⁻* recipients (Bestgene).

## Generation of the *REPTOR[WT]* allele

To generate a *REPTOR[WT]*, the sequence of *REPTOR* was amplified from genomic DNA of *w[1118]* flies using the following oligos: forward primer "CCGCGGCCGCCCCCTTCACCATGACAGAGAATCAGCTGTA" and reverse primer "GGGTCGGCGCGCCCACCCTTCATATAAAGCCCAGG CTCTT" and then cloned in a gateway-compatible vector by Gibson assembly reaction. The gene was then cloned in a *pBID-UAS-C* plasmid[80] using gateway cloning. Final plasmids were injected in fly embryos of the stock *yw; nos-Cas9 attP40* for targeted insertion in the second chromosome.

## Protein, triglyceride and glucose colorimetric measurements

Quantifications were done by adapting a published protocol[81]. Five or six biological replicates were collected from each cross at the designated time point. For each biological sample, eight thoraces were dissected in PBS by clipping the wings and the head, separating the abdomen from the thorax, and making sure the entire gut was removed. Dissected thoraces were immediately frozen on dry ice and stored at −80 °C until collections were finished. Samples were placed in an Abgene 96 Well 1.2 mL Polypropylene Deepwell Storage Plates (Thermo Fisher Scientific−AB0564) with 200 ul of ice-chilled PBS with 0.05% Triton-X-100 (Sigma) and Zirconium Oxide Beads, 1.0 mm, 1 Lb. (Next Advance Lab Products−ZROB10). Homogenization was done using a TissueLyser II homogenizer (QIAGEN), 2–3 times per 30 s with an oscillation frequency of 30 Hz/s. Plates were then centrifuged for 1 min at 3197 g to spin down debris and the homogenate was immediately used to quantify glucose and protein. For triglycerides, an additional step of heat treatment for 10 min at 75 °C was performed. Quantifications were done using a SpectraMax Paradigm Multi-mode microplate reader (Molecular Devices). For protein measurements, 5 μl of homogenate was diluted in 20 μl of PBS, mixed with 200 μl of BCA Protein Assay Kit (Pierce BCA Protein Assay Kit), and incubated for 30 min at 37 °C with gentle shaking in 96-well Microplates (Greiner Bio-One). Glucose was measured by mixing 10 μl of homogenate in 100 μl of Infinity Glucose Hexokinase Reagent (Thermo Fisher Scientific− TR15421), followed by incubation for 30 min at 37 °C in 96-well Microplates UV-Star (Greiner Bio-One – 655801) with gentle shaking. For triglyceride measurements, 20 μl of homogenate were used in 150 μl of Triglycerides Reagent (Thermo Fisher Scientific−TR22421) and incubated for 10 min at 37 °C in 96-well Microplates (Greiner Bio-One) with gentle shaking. Values were determined from standard curves of five serial dilutions (1:1) of BSA (Pierce BCA Protein Assay Kit), Glycerol standard solution (Sigma) or Glucose standard solution (Sigma).

## RNA extraction and qRT-PCR

*Drosophila* thoraces and guts: thoraces were dissected as described above. Five to ten thoraces and six to ten guts were collected per sample. Samples were homogenized in TRIzol reagent (Ambion) and mixed vigorously with Chloroform (Fisher) for phase separation. The collected supernatant was processed on Direct-zol RNA MicroPrep columns (Zymo Research) according to the manufacturer instructions. Eluted RNA was treated with TURBO DNA-free Kit (Ambion) and cleaned up again with the Direct-zol RNA MicroPrep columns. Reverse transcription was done by using iScript cDNA Synthesis Kit (Bio-Rad). qRT-PCR was done in a CFX96 Real-Time System (Bio-Rad) using iQ SYBR Green Supermix (Bio-Rad). Relative mRNA levels were calculated using the ΔΔCt method. For thoraces, values were normalized to the housekeeping gene *a-tubulin* or *rp49*. For guts, mRNA levels were normalized to *rp49*. Mouse C2C12 myotubes: Total RNA from myotubes was extracted by homogenizing samples in TRIzol followed by purification using RNeasy Mini spin columns (QIAGEN). Reverse transcription was performed using a High-Capacity cDNA Reverse Transcription kit (Applied Biosystems). qRT-PCR was done in a ABI PRISM 7900HT Real-Time PCR system (Applied Biosystems) using SYBR Green Supermix 2× qPCR master mix (Promega). Relative mRNA levels were normalized to *Tbp* mRNA and calculated using the ΔΔCt method. All primer sequences are listed in Supplementary Table 1.

## RNA-seq

*Drosophila* sample collection: For the time-course RNA-seq experiment, three time points were chosen for sample collection. Flies were incubated at 29 °C to induce gut *yki*-tumors, and thoraces were collected at 2 days and 14 days after tumor induction. For the tumor "shutdown" phase, after a period of tumor induction of 14 days, flies were shifted back to 18 °C and kept for an additional 24 days before thoraces were collected. For the *REPTOR[ACT]* overexpression dataset, gene induction was done for 4 days at 29 °C prior to sample collection. Library preparation for transcriptome analysis: Total RNA was quantified using an Agilent 2200 or 4200 TapeStation instrument (Agilent), with the corresponding Agilent TapeStation RNA assay (Agilent). The resulting RIN (RNA Integrity Number) scores, spanning between 8.8 and 9.6, and the respective concentrations of RNA were used to qualify samples for library preparation. For the time-course experiment on *Esg>yki[S3A]* thoraces, the rRNA was depleted using Illumina RiboZero HMR workflow. For the REPTOR and CREBRF overexpression experiments, the mRNA was pulled-down using oligo-dt beads as part of the KAPA mRNA HyperPrep workflow. Library samples were normalized in equimolar ratio and divided into four pools for technical replication. Each pool was denatured and loaded onto an Illumina NextSeq 500 instrument, with a High-Output 75-cycle kit to obtain Single-Read 75 bp reads. Transcriptome analysis—*Drosophila*: FastQC was used to assess the quality control of the sequencing data, and MultiQC was used to aggregate the results. Sequencing reads were then aligned to the *Drosophila* reference genome obtained from FlyBase (FB dmel r6.30) using STAR (v2.7.2b)[82]. Differentially expressed genes were identified using DESeq2 (v1.34.0). Final hits were selected based on both adjusted *P* value as well as fold change (FC) cut-off. In the time-course RNA-seq dataset, 510 genes showing differential expression between 2 days and 14 days after tumor induction were selected using a Log2FC > 1.3 cut-off. For the RNA-seq dataset of *REPTOR[ACT]* overexpression in muscle, 700 genes were selected using a Log2FC > 1 cut-off with adjusted *p* value < 0.05. The information on the differentially expressed genes for each dataset is presented in Supplementary Data 1, along with the gene ontology annotation and respective human orthologs identified by DIOPT (v9.2)[83]. The transcription factors differentially expressed in the time-course RNA-seq dataset were illustrated by heatmap in Supplementary Fig. 3h using TM4 software suite (http://mev.tm4.org/). The enrichment analysis was performed on differentially expressed genes of both datasets using PANGEA (v1), an in-house program based on hyper-geometric distribution[84]. The gene hits of each dataset were assembled based on their gene ontology (GO) annotation from Fly-Base, with a further selection of SLIM terms specifically for *Drosophila* genes. For the KEGG pathway annotation, the *Drosophila* genes were supplemented with the annotation for human genes identified using DIOPT with the high/moderate rank filter. Enrichment results for each dataset are presented in Supplementary Data 1. Transcriptome analysis —Mouse C2C12 myotubes: Sequenced reads were trimmed using fastp 0.20.1[85] and then aligned to the mm10 reference genome assembly using RNA STAR (v2.7.8)[82]. Differential gene expression testing was performed by analyzing gene counts from RNA STAR using DESeq2 (v2.11.40.6)[86]. Genes with mean expression below 0.5 were filtered, and the remaining genes were ranked by Wald statistic to generate a pre-ranked gene list for gene set enrichment analysis using GSEA (v4.0.1)[87,88]. 'Classic' mode was used for enrichment statistic

calculation. For unbiased discovery of enriched gene sets, the HALL-MARK gene sets were used as queries[89]. In a separate query, muscle gene expression data were examined using a custom gene set, 'PGC1α muscle gene set.', and the list of genes is presented in Supplementary Data 1. This gene set consists of 331 genes upregulated in the gastrocnemius muscle of MCK-PGC1α transgenic mice with a $p$ value < 1e-5[87]. $P$ values reported in figures correspond to FDR $q$ values to account for multiple hypothesis testing.

## snRNA-seq of thoraces
Adult males were incubated for 4 days at 29 °C to express either *REPTOR[ACT]* or *FoxO[ACT]* in muscle with *dMef2-GAL4*. Thoraces were dissected making sure the guts were completely removed and snap frozen. Single nuclei extractions were prepared as described previously with minor changes[90]. Briefly, thirty thoraces were homogenized in 1 ml of homogenization buffer[90] composed of 250 mM of sucrose (Invitrogen), 10 mM Tris pH 8.0, 25 mM KCl, 5 mM MgCl2, 0.1% Triton-X, 0.5% RNasin Plus (Promega - N2615), 1X protease inhibitor (Promega - G652A) and 0.1 mM DTT, using a 1 ml dounce (Wheaton − 357538). After filtering the extract with a 40 μm cell strainer (Falcon) and a 40 μm Flowmi (BelArt- H13680-0040), nuclei were collected via centrifugation for 10 min at 1000 g at 4 °C, and then re-suspended in 800 μl PBS/BSA buffer (1× PBS with 0.5% BSA and 0.5% RNasin Plus). The nuclei suspension was filtered again using a 40 μm Flowmi and stained with DRAQ7 Dye (Invitrogen, D15106). Single nuclei were sorted using a Sony SH800Z Cell Sorter. Around 400 thousand nuclei were collected, centrifuged at 1000 g for 10 min at 4 °C, and then resuspended at a final concentration of 800–1000 cells/μl in PBS/BSA buffer. Ten thousand nuclei per genotype, with two technical replicates per genotype, were encapsulated using 10x Genomics Chromium controller. After encapsulation, single-nuclei libraries were generated following the manufacturer instructions. Sequencing was done by NovaSEQ 6000 S2 using 1.65 billion reads in total, with over 30,000 reads obtained per nucleus per technical replicate. snRNA-seq data analysis. Raw data files were processed using Cell Ranger software (10x Genomics, v7.0.0) to generate the single cell matrix for each sample. All downstream analysis was performed in R (v4.1.2). The count matrices were then imported into Seurat (v4.3.0) and cells with <100 UMI or <100 genes detected were removed from the dataset. Batch correction was performed using the Harmony algorithm (v0.10)[91] before creating a UMAP and performing clustering analysis using a resolution of 0.3[92]. 45335 nuclei and 414 UMIs (mean) per nucleus were recovered for downstream analysis. The annotation of the clusters was done by comparing the top markers with the Fly Cell Atlas whole body_dm_FCAvs2_10x database using the DRSC-DB database[93]. Clusters that did not have an obvious annotation were considered "unknown". The default settings of Seurat (v4.3.0) were used to generate the dot plot of muscle specific markers portraying scaled expression. For the transcriptome of the four muscle clusters, differentially expressed genes were calculated using the Wilcox Rand Sum Test across each cluster and were then further selected using a Log2FC > 2 cut-off and adjusted $p$ value < 0.05 for downstream analysis. 10x Genomics Loupe Browser (v6.4.1) was used to generate the violin plots in Supplementary Fig. 5c. To screen for potential direct target genes of REPTOR, DNA binding sites for REPTOR/REPTOR-BP were identified based on chip-seq datasets using the modERN resource (https://epic.gs.washington.edu/modERN/)[94]. Information on the FoxO DNA binding sites was obtained from a relevant publication[95], as well as from the modERN database. Genes with DNA binding sites for REPTOR/REPTOR-BP or FoxO localized up to 10 kb upstream of the transcription start site (TSS) were listed in Supplementary Data 2. REPTOR and FoxO target genes: Differentially expressed genes detected in each of the four muscle clusters were screened to identify exclusive and common targets between REPTOR and FoxO in muscle (Supplementary Data 2). To validate the REPTOR exclusive hits and confirm that their regulation

could be detected in the mixed cell population of homogenized thoraces, we focused on seventeen genes whose expression was similarly altered in bulk RNA-seq of *dMef2 > REPTOR[ACT]* thoraces using a Log2FC > 1.5 cut-off. To validate the FoxO exclusive target genes and confirm that their regulation could be detected in homogenized thoraces, we focused on fifteen genes whose expression was also changed in bulk RNA-seq of *Esg>yki[S3A]* thoraces after 14 days of tumor induction using a Log2FC > 1 cut-off (Supplementary Data 1 "DEG Yki-thoraces − 14 days"). Only genes with predicted DNA binding sites for either REPTOR/REPTOR-BP or FoxO, and that were upregulated in more than one muscle cluster, were then considered for further analysis (Supplementary Data 2 "REPTOR final targets" and "FoxO final targets").

## Immunostaining of thoraces and guts
Thoraces were fixed for 30 min on a relaxing buffer[96] composed of PBS with 0.3% Triton-X-100 (PBT 0.3%), 5 mM MgCl2, 5 mM EGTA, 2 mM ATP and 4% formaldehyde (Polysciences). After fixation, thoraces were bisected with a Double Edge Razor Blade (Personna – 74-0002), washed three times in PBT 0.3% and then incubated in a blocking solution of PBT 0.3% with 5% BSA for 1 hour at room temperature. For gut immunostaining, dissected guts were fixed on PBT 0.1% with 4% formaldehyde for 20 min, washed three times in PBT 0.1% and then blocked in PBT 0.1% with 5% BSA for 1 h. Incubation with primary antibodies was done overnight at 4 °C in blocking solution. Antibodies used were mouse anti-CoxIV (1:200-1:500, Abcam – 33985), chicken anti-GFP (1:2000, Aves Labs−GFP-1020), mouse anti-bPS1 (1:50, DSHB−CF.6G11), mouse anti-dlg1 (1:50, DSHB – 4F3) and rabbit anti-amphiphysin[35] (1:200 – 9906). Corresponding secondary antibodies used were anti-mouse Alexa Fluor 488 (1:500, Thermo Fisher Scientific−A-21202), anti-chicken 488 (1:500, Jackson ImmunoResearch – 703-545-155) or Alexa Fluor 647 (1:500, Thermo Fisher Scientific−A-21244). For staining of myofibrils, Alexa Fluor 555 Phalloidin (1:200, Thermo Fisher Scientific) was incubated with secondary antibodies for 2 h at room temperature. Hemi-thoraces were mounted on u-Dish 35 mm high Glass Bottom (Ibidi) in VECTASHIELD Antifade Mounting Medium with DAPI (Vector Laboratories).

## Thorax cross-sections
To make agarose blocks, Intermediate Tissue-Tek Cryomolds (Sakura – 4566) were loaded with agarose 1% and cooled down. Several intact thoraces previously fixed in PBT 0.3% with 4% formaldehyde for 30 min were placed in the agarose before solidifying. Slices of the agarose blocks with approximately 300 μm of thickness were generated on a Vibratome (Leica VT 1000 M). Samples were immediately incubated in a blocking solution of PBT 0.3% with 5% BSA for 1 h at room temperature followed by immunostaining.

## Confocal imaging and analysis
Confocal imaging was performed using a fully motorized Ti2 inverted microscope equipped with a Yokogawa CSU-W1 Spinning disk scanhead with a single 50 μm pinhole disk, a PI 250 μm or Mad City Labs 500 μm Z piezo stage inserts, a TOPTICA iChrome MLE laser launch with directly modulated 405 nm 100 mW solid state, 488 nm solid-state, 561 nm DPSS and 640 nm solid state laser lines, and a SOLA SE Lumencor light engine for widefield illumination. Image acquisition was done using either a Plan Apo λ 60 × 1.4 NA immersion oil DIC (Figs. 1f, g, 4d−g; Supplementary Figs. 1a−d, 7b), a Plan Fluor 40×/1.3 NA immersion oil (Supplementary Figs. 1e, f, 2b, e, 4a), a 20×/0.75 N.A. Air DIC objective (Fig. 1d, e) or a Plan Apo 10x/0.45 Air DIC objective (Fig. 1a). Images were captured using an Andor Zyla 4.2 Plus sCMOS monochrome camera using the 16-bit dual gain digitizer mode and Nikon Elements Acquisition Software AR (v5.02). Signal from the different channels was acquired sequentially using a Semrock Di01-T405/488/568/647 multi-band pass dichroic mirror and band pass emission

filters for green (Chroma ET525/36 m), red (Chroma ET 605/52 m), far red (Chroma ET 705/72 m) channels and DAPI (Chroma ET455/50 m). Z stack acquisition was optimized for proper axial sampling, using a 0.3–0.6 μm step size for 60x, while acquiring all channels in the same focal plane to minimize axial shifts. For 40× objective lens (0.3 μm step size), 20× objective lens (0.9 μm step size) and 10× objective lens (5.6 μm step size) each channel was acquired sequentially. Image analysis: was done in Fiji/ImageJ (v2.9.0)[97]. For images of hemi-thoraces acquired with the 60x objective, maximum intensity projections were done using 10–20 stacks of each acquisition starting from the surface of the sample and a representative image was shown. For all other scans, the maximum intensity projections included all stacks. For cross-sections, either all stacks were included in the maximum projection (Supplementary Fig. 1e, f) or only 15 stacks (Supplementary Fig. 1e', f').

### Myofiber degradation analysis

For each experiment, up to 15 flies per vial were incubated at 29 °C and then thoraces were dissected at a defined time point, fixed for 40 min in relaxing buffer, bisected with a razor blade, and stained for 2 h with Alexa Fluor 555 Phalloidin (1:200, Thermo Fisher Scientific–A34055) at room temperature. Hemi-thoraces were mounted on u-Dish 35 mm high Glass Bottom (Ibidi) in VECTASHIELD Antifade Mounting Medium (Vector Laboratories). Z stack acquisition of samples on the Yokogawa CSU-W1 Spinning disk was done with the 20× objective. To score for myofiber degradation, each hemi-thorax was analyzed for changes in sarcomere structure that are observed either in *Esg>yki[S3A]* flies or when *REPTOR[ACT]* is overexpressed in muscle (see Fig. 1f, g; Supplementary Figs. 1a–d, 7b). Samples showing the sarcomere phenotype in at least one of the thoracic dorso-longitudinal muscles were scored as positive for displaying myofiber degradation. Hemi-thoraces in which it was not possible to confirm a wildtype morphology of the sarcomeres for all dorso-longitudinal muscles due to damage caused by dissections were not considered suitable for analysis.

### Targeted mass spectrometry analysis

Metabolite quantification in thoraces was adapted from a published protocol[98]. In detail, 45 dissected thoraces were dissected per genotype, snap-frozen in liquid nitrogen, and stored at −80 °C in extraction buffer (80% (v/v) aqueous methanol). Tissues were homogenized in 500 ml extraction buffer using 1 mm zirconium beads (Next Advance–ZROB10) in a Bullet Blender tissue homogenizer (model BBX24–Next Advance), for 10 min at 4 °C. An additional homogenization step using 250 ml of extraction buffer was performed. Metabolites were then pelleted by vacuum centrifugation. Pellets were re-suspended using 20 ml HPLC grade water for mass spectrometry. 5–7 μl were injected and analyzed using a hybrid 6500 QTRAP triple quadrupole mass spectrometer (AB/SCIEX) coupled to a Prominence UFLC HPLC system (Shimadzu) via selected reaction monitoring (SRM) of a total of 298 endogenous water-soluble metabolites for steady-state analyses of samples. Some metabolites were targeted in both positive and negative ion mode for a total of 309 SRM transitions using positive/negative ion polarity switching. ESI voltage was +4950 V in positive ion mode and −4500 V in negative ion mode. The dwell time was 3 ms per SRM transition and the total cycle time was 1.55 s. Approximately 9–12 data points were acquired per detected metabolite. Samples were delivered to the mass spectrometer via hydrophilic interaction chromatography (HILIC) using a 4.6 mm i.d x 10 cm Amide XBridge column (Waters) at 400 μl/min. Gradients were run starting from 85% buffer B (HPLC grade acetonitrile) to 42% B from 0–5 min; 42% B to 0% B from 5–16 min; 0% B was held from 16–24 min; 0% B to 85% B from 24–25 min; 85% B was held for 7 min to re-equilibrate the column. Buffer A was comprised of 20 mM ammonium hydroxide/20 mM ammonium acetate (pH = 9.0) in 95:5 water:acetonitrile. Peak areas from the total ion current for each metabolite SRM transition were integrated using MultiQuant v3.0 software (AB/SCIEX). Values for each

metabolite analyzed and the ID compound database are presented in Supplementary Data 4. Five biological replicates were analyzed per genotype.

### ATP quantification

ATP quantification was adapted from a published protocol[99]. Four biological replicates, each with 4 thoraces were homogenized in 100 μl of 5 M guanidine-HCl (Sigma) supplemented with 100 mM Tris pH 8.0 and 5 mM EDTA pH 7.8, immediately boiled for 3 min at 95 °C, and centrifuged at 15,700 g, at 4 °C, for 10 min to collect the supernatant. Extracts were used for quantification of luciferase as proxy for ATP content using the ENLITEN ATP Assay System Bioluminescence Detection Kit (Promega) according to the manufacturer instructions. Samples were diluted in water to fit the range of the standard curve (5 serial dilutions, 1:10). ATP levels were calculated from the standard curve by measuring luminescence (Relative Light Units) in a SpectraMax Paradigm Multi-mode microplate reader (Molecular Devices) using Corning 96-well Solid White Flat Bottom Polystyrene microplates (Costar). Values were normalized to protein content (Pierce BCA Protein Assay Kit).

### Data analysis, statistics, and reproducibility

GraphPad Prism (v9.5.1) was used for statistical analysis. For analysis of two groups, unpaired two-tailed *t*-test with Welch's correction was used to calculate the *p*-value ($p < 0.05$). For analysis of three groups, either two-way ANOVA or one-way ANOVA followed by Sidak multiple comparisons test was used to calculate the *p*-value ($p < 0.05$). For myofiber degradation analysis, two-tailed Fisher's exact test with a confidence interval of 95% ($p < 0.05$) was used for pairwise comparisons between two groups. For immunostainings, five animals were analyzed to confirm reproducibility in all samples and one representative image was shown. No data was excluded from analysis. No randomization or blinding was done during experiments and data analysis. No statistical method was used to predetermine sample size. Statistical analysis on the quantification of protein, glucose, triglyceride and ATP content in thoraces was done on pooled results from two independent experiments. Complete information on the *Drosophila* genotypes used for each figure are detailed in Supplementary Data 3.

### Western blot

*Drosophila thoraces*: To measure phosphorylated S6K and AKT protein levels, flies were kept at 29 °C in normal food supplemented with yeast paste. On the day of collection, flies were transferred to vials with agarose 0.8% supplemented with 2% sucrose (Fisher Scientific), maintained at 29 °C for 5–6 h, and then transferred back to normal food with yeast paste for 1–3 h before collection. Ten thoraces per genotype were dissected in PBS supplemented with 2.5× Complete, EDTA-free Protease Inhibitor (Roche) and 1× PhosSTOP phosphatase inhibitor (Roche). Thoraces were then homogenized in 1x Laemmli Buffer (Bio-Rad) supplemented with 355 mM 2-Mercaptoethanol (Sigma) and 5×-10× of Halt Protease and Phosphatase Inhibitor Cocktail (100×) (Thermo Fisher Scientific). Samples were immediately boiled at 90 °C for five minutes and centrifuged at 15,700 g, at 4 °C, for 10 min to recover the supernatant. To detect pS6K, pAKT, S6K, AKT and REPTOR, equal volumes of supernatant were used for each sample, with 30–40 μg of protein extract being loaded into wells. For tubulin detection, equal volumes of 10–15 μg of protein extract of each sample were loaded on the same gel. Samples were resolved on Mini Protein Gels Novex WedgeWell 8 to 16%, Tris-Glycine (Invitrogen) and transferred to Nitrocellulose Membranes (Bio-Rad). Membranes were blocked on a 5% dry nonfat milk solution with Tris-buffered saline with 0.1% TWEEN 20 (TBS-T) (Boston BioProducts) at room temperature for 30 min. Further incubation with primary antibodies was done overnight at 4 °C with gentle shaking either in TBS-T or TBS-T with 5% nonfat milk. After washing, membranes were incubated with HRP-

conjugated secondary antibodies at room temperature for 1 h and developed using a SuperSignal West Pico PLUS or SuperSignal West Femto Chemiluminescent Substrate (Thermo Fisher Scientific). HRP signal was visualized using ChemiDoc MP Imaging System (Bio-Rad) with automatic optimal exposure. Densitometry of the bands was performed using the Image Lab software (Bio-Rad v6.0.1) and calculated by determining the adjusted volume after background subtraction for each band of interest. Total S6K and total AKT were detected from the same membranes previously blotted for pS6K or pAKT, respectively. To achieve this, membranes were incubated in Restore PLUS Western Blot Stripping Buffer (Thermo Fisher Scientific) for 5 min at room temperature and then re-blotted with the respective primary antibodies. Primary antibodies used were rabbit anti-pAKT (1:1000, Cell Signaling – 4060), rabbit anti-Akt (1:1000, Cell Signaling – 9272), rabbit anti-pS6K (1:1000, Cell Signaling – 9209), guinea pig anti-S6K[100] (1:10000), mouse anti-tubulin (1:20000, Sigma−T5168), guinea pig anti-REPTOR[28] (1:1000). Secondary antibodies used were anti-mouse HRP (1:10000, Amersham−NXA931), anti-rabbit HRP (1:10000, Amersham−NA934) and anti-guinea pig HRP (1:10000, Jackson ImmunoResearch – 106-035-003). Mouse C2C12 myoblasts: Cell culture samples were prepared in ice-cold RIPA buffer (50 mM Tris pH 7.4, 150 mM NaCl, 1% NP40, 0.5% sodium deoxycholate, 0.1% SDS) supplemented with 1× Complete, EDTA-free Protease Inhibitor (Roche) and 1× PhosSTOP phosphatase inhibitor (Roche). Lysates were sonicated 7.5 min in a Diagenode water bath sonicator (high intensity; 30 s on, 30 s off cycles), centrifuged at 16,000 g for 15 min at 4 °C, and the supernatants were used for subsequent analyses. Protein concentration was determined using the BCA assay. Protein lysates were denatured in Laemmli buffer, resolved by 4–12% NuPAGE Bis-Tris SDS-PAGE (Invitrogen) and transferred to polyvinylidene difluoride (PVDF, 0.45 μm pore size) membrane. Primary antibodies were diluted in TBS containing 0.05% Tween-20, 5% BSA, and 0.05% NaN3. Primary antibodies used were: rabbit anti-TBP (1:1000, Cell Signaling – 44059) and mouse anti-PGC1α (1:1000, EMD Millipore−ST1202). Membranes were incubated overnight with primary antibodies at 4 °C. For secondary antibody incubation, anti-rabbit (Promega−W4011) or anti-mouse HRP (Promega−W4021) was diluted in TBS containing 0.05% Tween-20 and 5% dry nonfat milk. HRP signal was visualized using Crescendo Western HRP substrate (EMD Millipore) and an Amersham Imager 680. Detection of PGC1α required immunoprecipitation prior to immunoblotting. After preparation of RIPA lysates as described above, 0.5 mg of protein was immunoprecipitated using 5 μg anti-PGC1α antibody (Santa Cruz−sc518025) during overnight rotation at 4 °C. The next day, 20 μl of protein G agarose bead slurry (Thermo Fisher) was washed and added to each IP, followed by rotation at 4 °C for 2 h. Beads were washed 3× in RIPA buffer and immunoprecipitated protein was eluted in Laemmli buffer for analysis by immunoblot. Immunoblot was performed as above, except that the anti-PGC1α primary antibody was diluted in TBS with 0.05% Tween-20 and 5% nonfat dry milk during overnight incubation.

## Cell culture

Mouse C2C12 cells (ATCC−CRL-1772) and HEK293A cells (Thermo Fisher Scientific−R70507) were cultured in DMEM with 4.5 g/L glucose and 584 mg/L L-glutamine (Corning), supplemented with 10% sterile filtered fetal bovine serum (GeminiBio) and 1x penicillin/streptomycin (Gibco). Cells were maintained in 37 °C and 5% CO₂. Adenoviral vectors encoding *Gfp* or M. musculus *Crebrf* (Uniprot Q8CDG5-1) were constructed using the pAd/CMV/V5-DEST vector (Thermo Fisher). No epitope tag was added to the CREBRF polypeptide. Adenovirus was generated in HEK293A cells and titered using the Adeno-X Rapid Titer Kit (Takara Bio). For differentiation, C2C12 cells were grown to confluence in 6-well cell culture plates and washed with PBS, followed by addition of DMEM with 4.5 g/L glucose and 584 mg/L L-glutamine (Corning), supplemented with 2% horse serum (GemCell) and 1x

penicillin/streptomycin (Gibco). Medium was changed every two days thereafter. For viral transduction experiments, adenovirus was added to the medium for 3 h on day 4 of differentiation at 250 MOI in the presence of 5 μg/ml polybrene (Santa Cruz Biotechnology). Experiments were performed 3 days later, at day 7 of differentiation. To assess the effects of nutrient withdrawal, 6 well plates containing C2C12 myotubes differentiated to day 7 as above, were washed once with 3 ml PBS and then incubated in Hank's balanced salt solution without calcium or magnesium (Corning) for 4 h. To assess insulin signaling, C2C12 myotubes were differentiated to day 6 as above, washed with 3 ml PBS, and then incubated overnight (16 h) in DMEM without serum. Bovine pancreatic insulin (Sigma) was added to 100 nM for 10 min and then cells were harvested for immunoblot analysis. Mouse primary myoblasts: Primary myoblasts were isolated from female C57BL/6 J mice as previously described[101], and propagated on collagen-coated tissue culture plates (Corning) in growth medium containing an equal mixture of F10 (Thermo) and DMEM (Corning) supplemented with 20% fetal bovine serum (GeminiBio) and penicillin/streptomycin. One day prior to differentiation, myoblasts were plated on collagen-coated plates at a density of 60,000 cells per cm². The next day, differentiation was initiated by washing with PBS and adding differentiation medium: DMEM containing 5% horse serum (HyClone) and penicillin/streptomycin. One day later, differentiation medium was replaced and adenovirus encoding *Gfp* or *Crebrf* were added as above. Differentiation medium was replaced daily until cell harvest 3 days after transduction.

## Mice

Mice were housed at 23 °C and approximately 50% relative humidity under a 12 h light/dark cycle. Animals were fed a standard irradiated chow diet. All experiments used 8–12 week old male C57BL/6 J mice (Jackson Labs−stock number 000664). Animal experiments were performed according to procedures approved by the Institutional Animal Care and Use Committee (IACUC) of the Beth Israel Deaconess Medical Center.

## Cellular bioenergetic measurements in C2C12 myotubes

C2C12 cells were seeded in 24-well Seahorse XF V7-PS cell culture microplates and differentiated upon confluence, then transduced using adenovirus as described above. Three days after transduction, an ATP rate assay was performed. The cells were washed twice with 0.5 ml Seahorse XF DMEM medium pH 7.4 (Agilent) and then incubated in a CO2-free incubator at 37 °C for 45 min. At this point, the medium was removed and replaced with 0.5 ml Seahorse XF DMEM. Unless otherwise stated, the medium was supplemented with 10 mM glucose, 1 mM pyruvate, and 2 mM glutamine. Oxygen consumption and extracellular acidification were assessed in a standard ATP rate assay protocol in a XFe24 Seahorse Analyzer at 37 °C as follows: basal measurement (3 cycles), inject port A (3 cycles), inject port B (3 cycles). Injection port A contained oligomycin (1.5 μM final; Cell Signaling Technology) and injection port B contained rotenone/antimycin A (1 μM/1 μM final; Sigma). Each cycle consisted of 2 min 30 s mixing, 2 min waiting, and 3 min measuring. The contribution of respiration and glycolysis to ATP production were calculated using the Seahorse Analytics web application (Agilent). Measurements were normalized to per-well protein content as determined by BCA assay (Thermo Fisher Scientific). Experiments using primary myotubes were performed with the following changes. Primary myoblasts were plated at a density of 17,000 cells per well on 24-well Seahorse XF V7-PS cell culture microplates that were pre-coated with ECL cell attachment matrix (Sigma). The next day, differentiation was initiated, followed by adenoviral transduction using the timeframe described above.

## Glucose uptake assay

C2C12 or primary myotubes were differentiated in 12-well plates and transduced as described above. Three days after adenoviral transduction, the medium was removed and replaced with 0.25 ml PBS

containing 0.1 mM 2-deoxyglucose. The cells were incubated an additional 30 min at 24 °C and then lysed in stop buffer, allowing measurement of 2-deoxyglucose-6-phosphate using the Glucose Uptake-Glo assay (Promega) and a FLUOstar Omega luminescence plate reader (BMG Labtech).

### Lactate measurement

For C2C12 myotubes, DMEM medium with 2% horse serum was conditioned for 24 h by C2C12 myotubes transduced as above with adenovirus encoding *Gfp* or *Crebrf*. The medium was subsequently assessed for lactate content using a Lactate Colorimetric Assay Kit (Biovision). For primary myotubes, intracellular lactate was measured three days after adenoviral transduction using the Lactate-Glo assay (Promega) according to manufacturer's instructions. Luminescence was measured in a FLUOstar Omega plate reader (BMG Labtech).

### Glycogen measurement

Intracellular glycogen was measured three days after adenoviral transduction of C2C12 or primary myotubes using the Glycogen-Glo assay (Promega) according to manufacturer's instructions. Luminescence was measured in a FLUOstar Omega plate reader (BMG Labtech).

### Reporting summary

Further information on research design is available in the Nature Portfolio Reporting Summary linked to this article.

## Data availability

The data generated in this research is available in the Source Data file provided with this work. Reagents are available from the Lead Contact. The RNA-seq and snRNA-seq datasets generated in this work have been deposited in the Gene Expression Omnibus (GEO) databases under the following accession codes:

GSE189214, RNA-seq dataset of thoraces of wildtype flies or flies with gut *yki*-tumors at different time points after tumor induction.

GSE189218, RNA-seq dataset of thoraces upon overexpression of *REPTOR*[ACT] in muscle with *dMef2-GAL4*.

GSE228034, RNA-seq dataset of C2C12 myofibers overexpressing *Crebrf*.

GSE227038, snRNA-seq dataset of thoraces upon *REPTOR*[ACT] or *FoxO*[ACT] overexpression in muscle with *dMef2-GAL4*.

The snRNA-seq dataset has also been made publicly available at DRSC/TRiP Functional Genomics Resources data portal [https://www.flyrnai.org/scRNA/thorax/] to allow users to query the expression of any gene of interest in thoraces. Source data are provided with this paper.

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

## Acknowledgements

We are grateful to Bernard Mathey-Prevot, David Doupe and Jonathan Zirin for comments on the manuscript, John M. Asara and the BIDMC Mass Spectrometry Core for the LC/MS metabolic profiling services, Sudhir Gopal Tattikota for advice with the snRNA-seq experiments, Mikhail Kouzminov for helping with the snRNA-seq dataset analysis, Justin Bosch for sharing plasmids, Ben Ewen-Campen and Raghuvir Viswanatha for sharing qPCR primers, Aurelio Teleman for providing the REPTOR reagents, Andrew Zelhof for the anti-amphiphysin antibody, the TRiP at Harvard Medical School, BDSC and NIG for providing transgenic fly stocks used in this study, Frank Schnorrer, Maria Spletter and Nuno Luis for advice on thorax dissections, Christians Villalta for fly embryo injections, Yuan Feng for technical help with dissections and RNA extractions, Jenny Ro for technical help with the colorimetric assays, Paula Montero and the MicroN Core Facility for assistance with confocal microscopy, the BPF Next-Gen Sequencing Core Facility at Harvard Medical School for the RNA-sequencing services. P.S. was supported by a Human Frontiers Science Program long-term Fellowship LT000937/2016. This work is supported in part by the Cancer Grand Challenges partnership funded by Cancer Research UK (CGCATF-2021/100022) and the National Cancer Institute (1 OT2 CA278685-01). This work was also supported by NIH R01AR057352, P01CA120964 and R01NS101745 to NP, NIH grants R01DK123228 and R01DK119117 to B.M.S. and DRCRF Fellowship DRG-120-17 to P.A.D. N.P. is an investigator of HHMI. This article is subject to the new HHMI's Open Access to Publications policy. HHMI lab heads have previously granted a nonexclusive CC BY 4.0 license to the public and a sublicensable license to HHMI in their research articles. Pursuant to those licenses, the author-accepted manuscript of this article can be made freely available under a CC BY 4.0 license imme-diately upon publication.

## Author contributions

P.S. and N.P. designed the study in Drosophila. P.S. performed and analyzed the experiments with Drosophila with technical help from R.B. and E.F. P.S. and P.A.D. designed the experiments with mouse and C2C12 myotubes. P.A.D. performed and analyzed the Seahorse assays and the experiments with mouse and C2C12 myotubes with technical help from S.E.W. Y.H., H.W., J.R. and P.A.D. analyzed the RNA-seq datasets. Y.H. and W.C. analyzed the snRNA-seq datasets. Y.L. per-formed the flow cytometry to isolate nuclei for snRNA-seq. P.J. gener-ated the UAS-HA-FoxO reagents and provided technical support for the western blots. P.S., P.A.D. and N.P. wrote the manuscript. B.M.S. and P.J. provided critical feedback to the manuscript.

## Competing interests

The authors declare no competing interests.
