## [Peer Review File · Nature Communications]

REVIEWER COMMENTS

Reviewer #1 (Remarks to the Author):

This is an interesting manuscript looking at how the *Drosophila* transcription factor REPTOR, and its human homolog Crebrf, regulate cellular metabolism in muscle cells. The authors provide evidence that REPTOR/Crebrf promotes mitochondrial respiration and inhibits glycolysis. Since both of these factors are understudied, this work will be of interest to a wide audience of people interested in metabolic regulation, nutrient sensing and physiology.

I have two main suggestions which would strengthen the story for publication:

1. The data supporting an increase in REPTOR activity in the *yki* overexpressing flies could be strengthened. The transcriptional induction of REPTOR shown in Fig 2a seems very mild - less than 2-fold. It's unclear if this 2-fold increase is sufficient to lead to an increase in REPTOR protein, which would be needed to increase its activity, since mRNA levels and protein levels often do not correlate. Since REPTOR is also regulated via phosphorylation, there may be regulation occurring at this post-translational level. However the data supporting an increase in REPTOR activity are circumstantial. Some direct evidence showing either more REPTOR protein or more REPTOR activity would help. For instance, some options would be:

- a western blot on REPTOR protein levels from thoraxes of control or *yki* OE flies
- an activity assay using the published REPTOR unk-luciferase reporter
- a REPTOR western blot on cytosolic versus nuclear extracts of fly thoraxes to see if more REPTOR is in the nucleus
- Q-RT-PCR on a panel of REPTOR target genes

2. The P-S6K western blots shown in Suppl Fig 4A and in Fig 3i are of poor quality. These are used to claim that changes in REPTOR phosphorylation are not occurring, but it is hard to conclude anything about even S6K phosphorylation from these blots because the phospho-signal seems close to background. It seems likely that the phosphorylation is getting lost during sample preparation. I suggest re-trying these blots by crushing the fly samples directly in 1x Laemmli with 10x protease inhibitor and 5x phosphatase inhibitor, and immediately boiling. We've experienced that fly lysates have higher amounts of proteases and phosphatases compared to cell culture lysates (perhaps due to the presence of gut enzymes?) so elevated levels of inhibitors as well as rapid boiling are needed to get interpretable phospho-blots.

Minor Issues:

1. Suppl Fig 2A – the labeling “Akt protein levels” is perhaps not optimal since the main point of this panel is Akt phosphorylation, not total protein levels
2. Suppl Fig 2B – since 4EBP is also a target of REPTOR, which is induced in this system, the induction of 4EBP can't be used as evidence for increased FOXO activity. The authors can pick a panel of other FOXO targets that are not REPTOR targets. I believe there was a paper published by Linda Partridge's lab identifying FOXO targets in adult flies.
3. Line 120 should reference Suppl Fig 2g-h, not h-i
(also the following references to panels i and j are off)
4. Line 154-155: actually the reference cited [28] shows that REPTOR mainly regulates metabolism in pupae and adults, not larvae (opposite of what is written in lines 154-5)
5. Was pfk reported to be a REPTOR target gene? (ie is this a direct effect?)
6. I'm not sure if the data in Fig 5d-f are interpreted correctly. It seems unlikely that glucose uptake can be unaltered in Crebrf overexpressing cells (or even somewhat increased) with a concomitant drop by 50% in glycolysis, because all the glucose taken up must go somewhere - either glycolysis, or glycogen synthesis. So if glucose uptake stays the same and glycolysis drops by half, the cells should show strong glycogen accumulation. Is that the case? I gather from the M&M that glycolysis is quantified using extracellular acidification of the medium using the Seahorse instrument, which measures lactate secretion. But if Crebrf overexpressing cells funnel more of the glycolytic pyruvate into the TCA cycle instead of lactate, this will look like reduced 'glycolysis' even though it's actually just reduced lactate secretion and unaltered glycolysis. Could it be that a change in flux of pyruvate away from lactate secretion and towards the TCA cycle has occurred? What are the C2C12 cells using as a substrate for respiration (which increases) if not glucose – I would guess they are not doing much beta-oxidation. This could be figured out by doing respiration assays where different substrates are removed, or where beta-oxidation is blocked with etomoxir.

Reviewer #2 (Remarks to the Author):

The work performed by Saavedra et al. aims at showing that the transcription factor REPTOR is a key regulator of muscle metabolism and metabolic flexibility.

My major concerns rely to the physiological relevance of the different models which have been used to perform this study:

- C2C12 cell line is not a valid model to make metabolic flux measurements. All the results obtained with C2C12 myotubes have to be confirmed with primary muscle cells MACS sorted (with magnetic beads) or FACS sorted (with cell sorter).
- Analysis of glycolysis via qRT-PCR is not enough accurate, such as in Figure 2F. Glycolysis is a metabolic flux. Therefore, measurement of key enzyme activities and/or quantification of metabolites (such as pyruvate or lactate) are needed.
- even if the drosophila model of yorkie-induced gut tumors could be an interesting model, for example to find new pathways, showing the role of CREBRF in cachexia models (C26 mice model or APC mice) will reinforce the role of REPTOR and CREBRF in the muscle pathophysiology and metabolic flexibility.

Minor comments:

- lines 50-56 (introduction): these 3 sentences can be simplified with less repetitions.

Reviewer #3 (Remarks to the Author):

Summary

This paper utilises both an adult *Drosophila* model of gut tumours as well as mouse-derived cell lines to examine the effect of the *Drosophila* protein REPTOR and its mammalian ortholog CREBRF on metabolism in muscle. Gut *yki*-tumours cause shifts in the metabolism of muscles away from glycolysis and towards oxidative phosphorylation, and ultimately causing myofiber degradation. Decreasing insulin signalling is not sufficient to cause these phenotypes in a wildtype context, however, increasing signalling can ameliorate the effects. Using a time-course RNA-seq experiment, the authors identify REPTOR as being upregulated in tumour bearing flies, and using a binary expression system, find that muscle-specific knockdown of REPTOR ameliorates glucose and lipid metabolism defects seen in the

tumour bearing animal. The authors show that REPTOR on its own is sufficient to modulate glycolysis, and that REPTOR appears to play similar roles in starvation as it does in the tumour context. However, the authors cannot link the upregulation of REPTOR to the insulin pathway, suggesting another mode of regulation that is not further explored. Interestingly, knockdown of *Impl2* in the gut reduces muscle REPTOR expression, yet activation of insulin signalling in the muscle does not, suggesting that *Impl2* is rescuing this phenotype through another mechanism (it is also shown the mTOR pathway is not responsible for this). The combination of in vivo and in vitro *Drosophila* work, utilising metabolic and transcriptional readouts nicely shows that REPTOR plays a role in the metabolic flexibility of muscles. This data is then further supported by examination of the mammalian REPTOR orthologue, CREBRF, in cultured mouse C2C12 myotubes, showing that the phenotypes seen in the fly model are recapitulated in a mammalian model. The results ultimately show that the function of REPTOR, with concern to its effect on muscle metabolism flexibility, is conserved through its mammalian counterpart, CREBRF. Overall, this paper is a nice characterisation of how a protein influences muscle metabolism. It uses thorough methods for examining metabolic readouts, and combines multiple models to confirm results, especially across species. However, the role of insulin signalling in the muscles and its involvement in cachexia has been studied in Lee 2021 using the same *Yki* tumour model, so I feel there is not enough that is new for this paper to be considered for publication in Nature Comms.

Major comments:

1. Having REPTOR isolated on its own as a modulator of these phenotypes, but not actually finding out how it is regulated, or whether its proposed downstream targets actually contribute to the phenotypes leaves the story lacking mechanism.
2. It is not that surprising that the insulin pathway affects these metabolic pathways, but without it directly affecting REPTOR expression in the tumour context, I cannot see how the two are linked and therefore relevant to each other.
3. Experiments demonstrating the links between gut *Impl2* and the expression levels of REPTOR is needed.
4. Would REPTOR translate to other tumour types? The phenotype appears to be predominant muscle atrophy, maybe not detachment, quantification of thoraces myofiber degradation needs clarification, it is not described neither in figure legend or methods how this is done. How the cut off for counting something as myofiber degradation is determined is not clear.
5. Mechanism into what is activating REPTOR in the muscles in *Yki* tumour animals is not clear.
6. Some further investigation in the downstream targets of REPTOR, such as pfk, and other members of the metabolic chain, to see if they also rescue would be helpful.

Spelling, grammar, clarification

Line 322 "...as 'the' main energy..."

Line 357 Saying both REPTOR and CREBRF are functionally conserved makes them sounds like two different things, as opposed to homologues

Line 498 Brand name for the plate reader

Line 522 Start of this line needs clarification

Line 531 Brand name for tapestation.

Line 541 *Drosophila* needs to be in italics

Line 547 adjust should be adjusted

Line 549 TF should be transcription factor the first time

Line 609 'read' should be 'red'

Line 642 Say previous protocols or something, don't just end on 'from'

Line 659 missing some words at the start of a sentence

Line 725 why are some plasmids italicised and others are not

Line 788 Brand for plate reader

Minor Points

Line 97 A description of what was quantified for myofiber degradation would be appreciated. Someone not familiar with what degradation looks like would not understand what you have quantified.

Line 103 Could this circular morphology be a sign of muscle degradation leading to less pressure on the mitochondria as seen in (state that paper I looked at)

Line 141 A similar length control for the reduction in protein content would be nice

Figure 4 and Supp fig 5 Does ATP decrease in flies but increase in the cell model when REPTOR ACT is expressed?

Line 523 Should describe the TRIzol extraction 'as above' if that is what it is, otherwise give more details

Line 778 Normalised to protein content of each well?

Line 799 Should this be listed as a Table?

The whole RNA extraction and RT-PCR section sounds like it has been written by two different people, as there are inconsistencies in reference placement and tone. I suggest making this more consistent.

The whole RNA-seq section needs to be checked over for grammar and spelling mistakes. The *drosophila* section seems like it needs references, seeing as how many are in the mouse c2c12 cells section

Overall there is inconsistency with the use of spaces between values and their respective units. This needs to be addressed throughout

REVIEWER COMMENTS

Reviewer #1 (Remarks to the Author):

This is an interesting manuscript looking at how the *Drosophila* transcription factor REPTOR, and its human homolog Crebrf, regulate cellular metabolism in muscle cells. The authors provide evidence that REPTOR/Crebrf promotes mitochondrial respiration and inhibits glycolysis. Since both of these factors are understudied, this work will be of interest to a wide audience of people interested in metabolic regulation, nutrient sensing and physiology.

We thank the reviewer for appreciating our story on the function of REPTOR and CREBRF as regulators of muscle metabolism.

I have two main suggestions which would strengthen the story for publication:

1. The data supporting an increase in REPTOR activity in the *yki* overexpressing flies could be strengthened. The transcriptional induction of REPTOR shown in Fig 2a seems very mild - less than 2-fold. It's unclear if this 2-fold increase is sufficient to lead to an increase in REPTOR protein, which would be needed to increase its activity, since mRNA levels and protein levels often do not correlate. Since REPTOR is also regulated via phosphorylation, there may be regulation occurring at this post-translational level. However the data supporting an increase in REPTOR activity are circumstantial. Some direct evidence showing either more REPTOR protein or more REPTOR activity would help. For instance, some options would be:

- a western blot on REPTOR protein levels from thoraxes of control or *yki* OE flies
- an activity assay using the published REPTOR unk-luciferase reporter
- a REPTOR western blot on cytosolic versus nuclear extracts of fly thoraxes to see if more REPTOR is in the nucleus
- Q-RT-PCR on a panel of REPTOR target genes

We thank the reviewer for these critical suggestions. As any of them would significantly improve this manuscript, we attempted to address each of the four suggestions.

We agree that it is important to have more direct evidence supporting an increase in activity of REPTOR in *yki*-flies. We think that the most canonical way to show changes in REPTOR activity is by looking at specific target genes of REPTOR. However, in *yki*-flies, the transcription factor FoxO is most likely active as well due to systemic reduction of insulin signaling. Since REPTOR and FoxO share around 40% of target genes (PMID: 25920570, PMID: 18177722), it is not trivial to find exclusive targets for each transcription factor in muscle. Further, the bulk RNA-seq data of *REPTOR*^[ACT] gain-of-function in muscle presented in Supplementary Table 1 depicts the transcriptional changes occurring in whole thoraxes. In order to find novel, and specific, target genes of REPTOR in muscle that could be used as a readout of increased activity, we had to first isolate the muscle transcriptome when REPTOR is more active in this tissue. We therefore performed single nucleus RNA-seq (snRNA-seq) in extracted nuclei from thoraxes in which *REPTOR*^[ACT] was elevated with *dMef2-GAL4*. This allowed the identification of genes with differential expression specifically

in muscle clusters, while excluding the transcriptional contributions from other tissues. Next, we had to identify the potential target genes of REPTOR that were not regulated by FoxO. To perform this, we again used snRNA-seq to analyze the gene expression changes in muscle clusters while overexpressing an allele of *FoxO*^[ACT] with the *dMef2-Gal4* driver. This helped us exclude common target genes of the two transcription factors. By generating this dataset, we were able to identify four distinct muscle clusters and isolate the transcriptional profile of muscle tissue when *REPTOR*^[ACT] or *FoxO*^[ACT] were overexpressed in muscle (new Figure 3). We discovered a total of 197 REPTOR-specific target genes, 120 FoxO-specific target genes and 63 common target genes with differential expression in muscle clusters. The results of this experiment generated a new main Figure 3, and two associated new Supplementary Figures 5 and 8, in this revised version. We have also rewritten the methods section.

Our subsequent analysis, which included informatic analysis of putative REPTOR/REPTOR-BP DNA binding sites (see new Methods section on snRNA-seq), identified *CG42390* and *RhoGAP68F* as targets of REPTOR in muscle. These genes were increased in thoraces of *yki*-flies in a manner dependent on *REPTOR* expression in muscle tissue (new Figure 3e, f). Thus, these results support the hypothesis that REPTOR activity is increased in the muscle tissue of *yki*-flies.

We also performed an important experiment to support our hypothesis that increased activity of REPTOR in muscle is sufficient to drive the myofiber degradation and glucose content increase in *yki*-flies. We did this by investigating whether elevating REPTOR protein levels in muscle using a wildtype allele, while simultaneously repressing the mTOR pathway by co-expressing PRAS40 or Tsc1/Tsc2, could phenocopy the muscle degradation and glucose content increase observed in *yki*-flies. We were surprised to discover that the *UAS-REPTOR*^[WT] allele (PMID: 25920570) that we previously used was not elevating the protein levels of REPTOR significantly when overexpressed with the *dMef2-GAL4* driver. Given that this line was shown to induce the expression of several target genes (PMID: 25920570), we concluded that our copy of this stock may have suffered some loss-of-function mutation.

We generated a new *REPTOR*^[WT] allele, which we have now included in the methods section and validated by western blot (see new Figure 4a). We replaced any results using the previous allele of *REPTOR*^[WT] with the new data obtained with our new allele. Notably, the simultaneous overexpression in muscle of our new *REPTOR*^[WT] allele with *PRAS40* or *Tsc1/Tsc2* increased the myofiber degradation and glucose content, similar to the phenotypes observed in *yki*-flies (see new Figure 4b, c). This contrasts with the lack of effect of a single overexpression of *REPTOR*^[WT], and of the mTOR inhibitors PRAS40 or Tsc1/Tsc2 in muscle (see new Supplementary Figure 6). These new results strongly support our hypothesis that, in *yki*-flies, REPTOR sustained activity promotes the myofiber degradation and glucose content increase. Given the importance of these results, we generated a new Figure 4. Also, we moved the previous results showing the impact of *REPTOR*^[ACT] overexpression in muscle to a Supplementary Figure 7, as they are just a confirmation of these new results obtained with *REPTOR*^[WT].

Of note, regarding the other options suggested by the reviewer to show evidence of increased REPTOR activity, we explored option 2 and looked whether *REPTOR*^[ACT] gain-of-function in muscle could increase the *unkempt* target gene expression. Neither the thoracic bulk RNA-seq (Supplementary Table 1) nor the snRNA-seq dataset (Supplementary Table

2) showed a significant increase in expression of *unkempt*, suggesting that it might not be induced by REPTOR in muscle.

We also addressed option 1 and attempted to detect changes in protein levels of REPTOR in thoraces of *yki*-flies using a published antibody kindly shared by Aurelio Teleman group (PMID: 25920570). Yet, we could only observe a modest increase in protein levels (see new Figure 2c). Based on the Fly Cell Atlas database (PMID: 35239393), *REPTOR* is expressed in other tissues, which may cause a “diluting effect” that masks a potential increase of REPTOR protein levels in muscle tissue alone. The technical challenges of performing option 2 directly relate to option 3, in which detecting changes in REPTOR levels in cytosolic and nuclear fraction would be likely even harder to achieve. We were also informed that the REPTOR antibody is not suitable for immunostaining. As such, given the technical challenges and the lack of suitable reagents, we concluded that it would not be possible to convincingly determine the subcellular location of REPTOR in the muscle *yki*-flies.

2. The P-S6K western blots shown in Suppl Fig 4A and in Fig 3i are of poor quality. These are used to claim that changes in REPTOR phosphorylation are not occurring, but it is hard to conclude anything about even S6K phosphorylation from these blots because the phospho-signal seems close to background. It seems likely that the phosphorylation is getting lost during sample preparation. I suggest re-trying these blots by crushing the fly samples directly in 1x Laemmli with 10x protease inhibitor and 5x phosphatase inhibitor, and immediately boiling. We’ve experienced that fly lysates have higher amounts of proteases and phosphatases compared to cell culture lysates (perhaps due to the presence of gut enzymes?) so elevated levels of inhibitors as well as rapid boiling are needed to get interpretable phospho-blots.

We thank the reviewer for this remark on the western blots. We have repeated all western blots following the reviewer’s suggestion and replaced the previous results with new immunoblots in this revised version. In detail, we dissected thoraces in PBS supplemented with 2x protease inhibitors and 1x phosphatase inhibitors, homogenized the samples in 1x Laemmli buffer with 5x Halt cocktail of protease and phosphatase inhibitors and immediately boiled the samples. However, we noticed that the overall signal of pS6K was improved if *yki*-flies were also incubated on fly food supplemented with yeast paste. Importantly, this did not affect the myofiber degradation or blunt the glucose content increase in thoraces of *yki*-flies. In addition, we introduced a synchronization step, in which we starved the flies for 5-6 hours, on the day of collection, and then transferred them back to fly food with yeast paste for 1-3 hours before collecting the samples. This helped improving the reproducibility of the results. We have rewritten the methods accordingly.

Following this new protocol, we were now able to detect a reduction in the ratio of pS6K/S6K levels in thoraces of flies bearing gut *yki*-tumors (see new Figure 2b). Consistent with previous studies showing that insulin signaling positively regulates mTOR activity, we also observed a rescue of this ratio when *ImpL2* was knocked down in gut *yki*-tumors (see new Figure 2b), and an increase in the ratio when *Myr-AKT* was overexpressed in muscle of *yki*-flies (see new Supplementary Figure 5e). Of note, some samples of *yki*-thoraces exhibited a reduction in the levels of total S6K (see new Supplementary Figure 5e), which suggests a very dynamic regulation of the amount of S6K protein in the context of gut *yki*-

tumors. Nevertheless, our new immunoblots showing a decreased ratio of pS6K/S6K *yki*-flies strongly support the hypothesis that REPTOR is more active in the muscle of these flies.

Minor Issues:

1. Suppl Fig 2A – the labeling “Akt protein levels” is perhaps not optimal since the main point of this panel is Akt phosphorylation, not total protein levels

We have changed the title of this western blot to “Immunoblot of AKT phosphorylation status”. The panel was replaced by a new western blot picture but remains as Supplementary Figure 2A.

2. Suppl Fig 2B – since 4EBP is also a target of REPTOR, which is induced in this system, the induction of 4EBP can't be used as evidence for increased FOXO activity. The authors can pick a panel of other FOXO targets that are not REPTOR targets. I believe there was a paper published by Linda Partridge's lab identifying FOXO targets in adult flies.

We thank the reviewer for this comment. This is an important point and we agree that *4E-BP* cannot be used as evidence of activity for REPTOR or FoxO, since it is a target of both transcription factors. We therefore edited the figures related to *4E-BP* expression in this revised version. Given the reviewer's comment, we decided to leave out the panels of results measuring *4E-BP* mRNA levels in context of *Esg>Yki^{S3A}* thoraces as these would just create confusion when assessing the likely higher activity of REPTOR or FoxO in flies bearing gut *yki*-tumors. We have only included results on *4E-BP* expression to validate the effect of overexpressing REPTOR or *FoxO* in muscle with *dMef2-GAL4*. Thus, in the new Supplementary Figure 5, which is associated with the snRNA-seq dataset (new Figure 3), we show an increase in *4E-BP* mRNA levels in thoraces when *REPTOR* or *FoxO* are overexpressed in muscle (Supplementary Figure 5a, b), and a third panel (new Supplementary Figure 5c) showing the violin plots of *4E-BP* expression for each of the muscle clusters of our snRNA-seq.

To search for FoxO target genes in muscle in the context of *yki*-tumors, the reviewer suggested we looked at a relevant paper from Linda Partridge's lab (PMID:21694719) that identified multiple target genes in adult flies. However, this still presents the problem of finding specific targets of FoxO in muscle of *yki*-flies that are not regulated by REPTOR. Instead, we used our snRNA-seq analysis to check for FoxO potential target genes in muscle, and whose expression was also changed in the bulk RNA-seq dataset of thoraces of *yki*-flies. By using informatic analysis to find which targets had putative FoxO DNA binding sites (see Methods section), we identified two potential targets of FoxO in muscle, *CG10383* and *I(1)G0469*, that were upregulated in thoraces of *yki*-flies, therefore suggesting an increase in the activity of FoxO. These results were included in a new Supplementary Figure 8, and the methods section has been updated.

3. Line 120 should reference Suppl Fig 2g-h, not h-l (also the following references to panels i and j are off)

We apologize for this. However, the link between insulin signaling and REPTOR has been extensively rewritten to answer a remark from reviewer 3. These results were rearranged,

and the panel showing the thoracic glucose content in the *dMef2>p60* background was moved to a new Supplementary Figure 9e.

4. Line 154-155: actually the reference cited [28] shows that REPTOR mainly regulates metabolism in pupae and adults, not larvae (opposite of what is written in lines 154-5)

This sentence has been rewritten.

5. Was *pfk* reported to be a REPTOR target gene? (ie is this a direct effect?)

Our analysis by snRNA-seq indicates that the expression of *pfk* is reduced in the indirect flight muscle cluster 0 when *REPTOR^{ACT1}* is overexpressed in muscle (see new Supplementary Table 2). Also, we identified putative binding sites for REPTOR/REPTOR-BP in the upstream region to the promoter (see methods section). As such, these observations suggest that the effect of REPTOR in decreasing *pfk* expression in muscle may be direct.

6. I'm not sure if the data in Fig 5d-f are interpreted correctly. It seems unlikely that glucose uptake can be unaltered in *Crebrf* overexpressing cells (or even somewhat increased) with a concomitant drop by 50% in glycolysis, because all the glucose taken up must go somewhere - either glycolysis, or glycogen synthesis. So if glucose uptake stays the same and glycolysis drops by half, the cells should show strong glycogen accumulation. Is that the case?

We have clarified this point with new experiments. In our original submission, the glucose uptake assay was not meant to be directly compared to our Seahorse data because they were performed under different conditions: the former was designed to test insulin action, and so involved extended serum starvation. In this condition, in which endogenous CREBRF activity is expected to be heightened, the impact of *Crebrf* overexpression may be less pronounced.

Therefore, we have now replaced these data with new glucose uptake and glycogen content assays in both primary myotubes as well as C2C12 myotubes. In all of these experiments, we do not perform serum starvation pretreatments, thus better matching the conditions of our Seahorse metabolic flux assays and allowing for comparison. In both cell types, forced *Crebrf* expression reduces glucose uptake (new Figures 6f and new Supplementary Figure 10e). In addition, in both cell types, forced *Crebrf* expression causes a mild accumulation of intracellular glycogen (new Figures 6g and new Supplementary Figure 10f). These findings suggest that reduced availability of glucose (via reduced cellular import and/or reduced glycogenolysis) contributes to the reduced glycolytic ATP production observed upon *Crebrf* expression.

I gather from the M&M that glycolysis is quantified using extracellular acidification of the medium using the Seahorse instrument, which measures lactate secretion. But if *Crebrf* overexpressing cells funnel more of the glycolytic pyruvate into the TCA cycle instead of lactate, this will look like reduced 'glycolysis' even though it's actually just reduced lactate secretion and unaltered glycolysis. Could it be that a change in flux of pyruvate away from lactate secretion and towards the TCA cycle has occurred? What are the C2C12 cells using as a substrate for respiration (which increases) if not glucose – I would guess they are

not doing much beta-oxidation. This could be figured out by doing respiration assays where different substrates are removed, or where beta-oxidation is blocked with etomoxir.

One strength of our study is that it does not rely solely on extracellular acidification to measure glycolysis. Extracellular acidification is confounded by the fact that mitochondrial respiration releases CO₂ into the media, which hydrates to H₂CO₃ and then dissociates to HCO₃⁻ and H⁺, thereby causing media acidification (PMID: 25449966). For this reason, our study relies on the ATP rate assay, a recently-developed approach in which simultaneous measurement of oxygen consumption and extracellular acidification in a medium of known buffering capacity allows for correction of the CO₂ effect. This enables delineation, in a single experiment, of the distinct contributions of glycolysis versus mitochondrial respiration to total cellular ATP production (PMID: 28270511). This assay is particularly well-suited to study the effects of genes like *Crebrf* that influence the balance of glycolysis and respiration.

In the ATP rate assay, mitochondrial ATP production is defined as the ATP production that relies on ATP synthase (and is thus inhibitable by oligomycin). This includes mitochondrial oxidation of pyruvate generated from glycolysis, as well as mitochondrial use of NADH reducing equivalents generated by glycolysis. In contrast, glycolytic ATP production is measured as the acidification caused by export of lactate/H⁺ into the media. As the Reviewer points out, 'glycolytic ATP production' defined in this way is not always equivalent to flux in the upstream steps of glycolysis: one can imagine a scenario in which increased mitochondrial demand for pyruvate oxidation leads to less lactate/H⁺ export and thus less 'glycolytic ATP production' despite high flux through the early steps of glycolysis.

To minimize the likelihood of such a scenario (in which interpretation of glycolytic ATP production is challenging), we perform our ATP rate assays in defined conditions of energy substrate availability. In most cases, not only glucose (10mM) but also pyruvate (1mM) and glutamine (2mM) are available for cellular respiration. Thus, if mitochondrial demand increases upon *Crebrf* expression, pyruvate is readily available, and need not be drawn from products of glycolysis.

We favor the interpretation that reduced glycolytic ATP production is due in substantial part to reduced flux in the upstream steps of glycolysis. This is based on several observations taken together. First, as described above, we observe the reduction in glycolytic ATP production not only in conditions of no exogenous energy substrate (Supplementary Figure 10b), but also in the presence of abundant exogenous pyruvate and glutamine, a setting in which mitochondrial respiration is not dependent on glycolysis-derived products (Figure 6d). Second, we observe reduced glucose uptake and increased intracellular glycogen stores in the setting of forced *Crebrf* expression (new Figure 6f, g). Third, we observe decreased expression of glycolysis-associated genes in the context of forced *Crebrf* expression (Supplementary Figure 10i).

Future work, especially in vivo, will be important to determine CREBRF's effects on lipid oxidation. This has been complicated thus far by the fact that the effects of CREBRF on the ATP rate assay occur even in the absence of exogenous energy substrates in the media, making it difficult to assess fuel use via fuel drop-out studies. In the context of media with glucose, pyruvate, and glutamine, we have found that etomoxir does not appear to have a

proportionally greater effect on oxidation in cells expressing *Crebrf*, suggesting that at least in this context, CREBRF does not necessitate lipid oxidation for mitochondrial fuel supply.

Reviewer #2 (Remarks to the Author):

The work performed by Saavedra et al. aims at showing that the transcription factor REPTOR is a key regulator of muscle metabolism and metabolic flexibility.

My major concerns rely to the physiological relevance of the different models which have been used to perform this study:

- C2C12 cell line is not a valid model to make metabolic flux measurements. All the results obtained with C2C12 myotubes have to be confirmed with primary muscle cells MACS sorted (with magnetic beads) or FACS sorted (with cell sorter).

C2C12 myoblasts (and their corresponding myotubes) have been a mainstay of studies of cellular metabolism for decades owing to their genetic tractability and consistent behavior in culture (PMID: 32812252). In particular, studies of insulin sensitivity, exercise-like stimuli, and atrophy have often utilized the C2C12 model. Furthermore, the ATP rate assay utilized in our studies was initially validated in C2C12 cells (PMID: 28270511). Nevertheless, a recent study enumerated significant metabolic and gene expression differences amongst the common *in vitro* muscle models (C2C12, L6, and primary cells), which highlights the danger of making universal conclusions from the study of any single system (PMID: 31825657).

Therefore, we have heeded the reviewer's advice and performed additional metabolic experiments in primary mouse myotubes. As in C2C12 myotubes, forced expression of *Crebrf* in primary myotubes caused a significant reduction in glycolytic ATP production, resulting in an overall shift towards mitochondrial oxidative metabolism, as measured using the Seahorse bioanalyzer (new Supplementary Figure 10c). This change was associated with decreased intracellular lactate (new Supplementary Figure 10d), decreased glucose uptake (new Supplementary Figure 10e), and mildly increased glycogen stores (new Supplementary Figure 10f). These results are consistent with our findings in C2C12 myotubes and thus strengthen the generality of our conclusion that CREBRF activity reduces cellular preference for glucose energy metabolism.

- Analysis of glycolysis via qRT-PCR is not enough accurate, such as in Figure 2F. Glycolysis is a metabolic flux. Therefore, measurement of key enzyme activities and/or quantification of metabolites (such as pyruvate or lactate) are needed.

To address this remark, we have investigated the levels of glycolytic and TCA cycle metabolites by LC/MS analysis in thoraces with *REPTOR^[ACT]* overexpression in muscle. The reviewer suggested looking at lactate levels. However, there is strong evidence that glycolysis is completely aerobic in insects (PMID: 5908129) and that the activity of lactate dehydrogenase is significantly low in thoraces (PMID: 13428988, PMID: 4342385). Therefore, we do not think that changes in levels of lactate can be used reliably as a readout of glycolytic flux in *Drosophila*. Instead, we looked at other metabolites and confirmed a significant reduction of the first four glycolytic metabolites (new Figure 5a), which supports our hypothesis that REPTOR is a repressor of glycolysis in muscle. We also

co-expressed *REPTOR*^[ACT] in muscle with *Myr-AKT* to increase insulin signaling and induce glycolysis, and observed an amelioration of the myofiber degradation (new Figure 5d). This was consistent with our previous results in *yki*-flies, in which the induction of insulin signaling in muscle ameliorated the myofiber degradation (see new Figure 1i). These new results, therefore, suggest that REPTOR sustained activation in muscle collapses the energy metabolism program by repressing glycolysis.

Furthermore, the LC/MS analysis revealed that several TCA cycle metabolites were actually increased when *REPTOR*^[ACT] was overexpressed in muscle (see new Figure 5), raising the possibility of TCA cycle stimulation. Importantly, muscle-specific knockdown of *CPT1*, the fatty acid importer to mitochondria, in either flies bearing gut *yki*-tumors (see new Figure 1j) or when co-expressed with *REPTOR*^[ACT] in wildtype muscle (see new Figure 5f) ameliorated the myofiber degradation. Thus, our new results suggest that REPTOR sustained activity may also increase the fatty acid import to mitochondria.

In the previous version of this manuscript, we included a Seahorse analysis of S2R+ cells overexpressing *REPTOR*^[ACT]. However, in this *in vitro* model, REPTOR did not affect the glycolytic contribution to ATP production, even though it increased the oxygen consumption rate and the mitochondrial contribution to ATP production. Given the reviewer's suggestion to look at levels of glycolysis and TCA cycle metabolites, we think that this new LC/MS analysis *in vivo* in thoraces is more relevant than the data obtained from S2R+ cells. As such, we have replaced the S2R+ cells metabolic analysis with the LC/MS analysis *in vivo* in thoraces.

- even if the drosophila model of yorkie-induced gut tumors could be an interesting model, for example to find new pathways, showing the role of CREBRF in cachexia models (C26 mice model or APC mice) will reinforce the role of REPTOR and CREBRF in the muscle pathophysiology and metabolic flexibility.

To address the potential role of CREBRF in the context of mammalian cachexia, we examined *Crebrf* expression in a relevant mouse model. In publicly available RNA-seq datasets (PMID: 35941104), *Crebrf* mRNA in the gastrocnemius muscle of mice with cancer anorexia cachexia syndrome in a genetic model of non-small cell lung cancer was elevated 64% as compared to its level in mice with lung tumors but without cachexia (new Supplementary Figure 10a). These findings suggest that CREBRF activity is increased in at least some settings of mammalian cachexia.

To experimentally address the effect of elevated *Crebrf* expression in muscle, we injected AAV encoding *Crebrf* or *Gfp* (driven by a CMV promoter) into the hindlimb muscles of wild-type adult mice. Although the *Gfp* transcript was expressed at high levels, viral *Crebrf* expression did not overcome the endogenous transcript levels, such that the total levels of *Crebrf* transcript in tibialis anterior and gastrocnemius muscles were unchanged. Our experiments have therefore been unable to address the effect of elevated *Crebrf* expression in either wild-type or tumor-bearing mice. Future studies using genetic models of *Crebrf* expression in mice will likely be needed to address this question.

Minor comments:

- lines 50-56 (introduction): these 3 sentences can be simplified with less repetitions.

We have fixed it.

Reviewer #3 (Remarks to the Author):

Summary

This paper utilises both an adult *Drosophila* model of gut tumours as well as mouse-derived cell lines to examine the effect of the *Drosophila* protein REPTOR and its mammalian ortholog CREBRF on metabolism in muscle. Gut *yki*-tumours cause shifts in the metabolism of muscles away from glycolysis and towards oxidative phosphorylation, and ultimately causing myofiber degradation. Decreasing insulin signalling is not sufficient to cause these phenotypes in a wildtype context, however, increasing signalling can ameliorate the effects. Using a time-course RNA-seq experiment, the authors identify REPTOR as being upregulated in tumour bearing flies, and using a binary expression system, find that muscle-specific knockdown of REPTOR ameliorates glucose and lipid metabolism defects seen in the tumour bearing animal. The authors show that REPTOR on its own is sufficient to modulate glycolysis, and that REPTOR appears to play similar roles in starvation as it does in the tumour context.

However, the authors cannot link the upregulation of REPTOR to the insulin pathway, suggesting another mode of regulation that is not further explored. Interestingly, knockdown of *Impl2* in the gut reduces muscle REPTOR expression, yet activation of insulin signalling in the muscle does not, suggesting that *Impl2* is rescuing this phenotype through another mechanism (it is also shown the mTOR pathway is not responsible for this). The combination of in vivo and in vitro *Drosophila* work, utilising metabolic and transcriptional readouts nicely shows that REPTOR plays a role in the metabolic flexibility of muscles. This data is then further supported by examination of the mammalian REPTOR orthologue, CREBRF, in cultured mouse C2C12 myotubes, showing that the phenotypes seen in the fly model are recapitulated in a mammalian model. The results ultimately show that the function of REPTOR, with concern to its effect on muscle metabolism flexibility, is conserved through its mammalian

counterpart, CREBRF. Overall, this paper is a nice characterisation of how a protein influences muscle metabolism. It uses thorough methods for examining metabolic readouts, and combines multiple models to confirm results, especially across species. However, the role of insulin signalling in the muscles and its involvement in cachexia has been studied in Lee 2021 using the same *Yki* tumour model, so I feel there is not enough that is new for this paper to be considered for publication in Nature Comms.

We thank the reviewer for the comments on our work. However, we would also take the opportunity to clarify that this study focuses on the discovery and characterization of REPTOR and CREBRF as repressors of glycolysis and as regulators of energy metabolism, and not on the role of insulin signaling in the *yki*-flies. In fact, we showed that reducing insulin signaling systemically or specifically in muscle tissue of wildtype flies was not sufficient to induce the phenotypes observed in *yki*-flies or when *REPTOR*^[ACT] was overexpressed in muscles (see new Supplementary Figure 9d, i).

In addition, our mammalian work on CREBRF, including the global characterization of the gene expression program that it regulates and the experimental demonstration of its conserved function in repressing glycolysis and promoting OXPHOS, are novel and of general interest.

As such, our work significantly departs from previous published work, in particular *Lee et al, 2021* (PMID: 34078667), and we kindly disagree with the reviewer's statement that this manuscript does not have enough novelty.

Major comments:

1. Having REPTOR isolated on its own as a modulator of these phenotypes, but not actually finding out how it is regulated, or whether its proposed downstream targets actually contribute to the phenotypes leaves the story lacking mechanism.

We thank the reviewer for this comment and agree that both the regulation and the mechanism of action of REPTOR in muscle are important to address. For this revised manuscript, we generated relevant new data focusing on both aspects.

Regarding the regulation of REPTOR, we have generated new western blots using a more suitable protocol for extraction of phosphorylated proteins from adult fly thoraces. We confirmed that thoraces of flies with gut *yki*-tumors have indeed a lower ratio of pS6K/S6K, an indicator of reduced mTOR activity. These results strongly suggest that REPTOR activity is increased in flies bearing gut *yki*-tumors and were included in a new Figure 2b. We have also shown that co-expression of *REPTOR*^{WT} with pRAS40 or Tsc1/Tsc2, two repressors of the mTOR pathway, induced the increase in glucose content and myofiber degradation, similarly to the overexpression of the constitutively active allele *REPTOR*^{ACT}. These results were included in a new Figure 4 and demonstrate that mTOR represses REPTOR *in vivo* in muscle, and that the sustained activation of REPTOR leads to muscle degradation and impairment of glucose metabolism.

Regarding the mechanism of action of REPTOR in muscle, our initial work suggested that REPTOR is a repressor of glycolysis while potentially promoting the use of other energy substrate like lipids. As such, we hypothesized that the sustained activity of REPTOR in muscle led to the disruption of the energy metabolism program and myofiber degradation due to the likely preference of flight muscles for using glucose (PMID: 15395188).

In this revised version, we have edited Figure 1 to include the results showing an amelioration of myofiber degradation of *yki*-flies when insulin signaling is increased in muscle, which promotes glycolysis (see new Figure 1i and new Supplementary Figure 2c, d). We also observed that knocking-down *CPT1*, the fatty acid importer into mitochondria, in the muscle of *yki*-flies, ameliorated the myofiber degradation (see new Figure 1j). We then confirmed that these same genetic manipulations (increasing insulin signaling or knocking-down of *CPT1* in muscle) led to a similar amelioration of the myofiber degradation phenotype caused by overexpressing *REPTOR*^{ACT} (see new Figure 5d, f).

As such, these new results provide a mechanistic insight into the function of REPTOR in regulating metabolism. Importantly, they support our hypothesis that REPTOR induces myofiber degradation by repressing glycolysis and likely favoring fatty acids as an energy substrate, which may then become deleterious to the muscle.

2. It is not that surprising that the insulin pathway affects these metabolic pathways, but

without it directly affecting REPTOR expression in the tumour context, I cannot see how the two are linked and therefore relevant to each other.

First, we would like to clarify that the knockdown of *ImpL2* in tumors blunts the upregulation of *REPTOR* in thoraces, while also preventing the increase in glucose content and myofiber degradation caused by gut *yki*-tumors. These two latter phenotypes could be recapitulated by overexpression of *REPTOR*^[ACT] in muscle. Furthermore, the knockdown of *REPTOR* in muscle of *yki*-flies restores *REPTOR* expression to that of non-tumor-bearing flies and blunts the increase in glucose content. These results therefore indicate a link between *ImpL2* and *REPTOR* in the context of gut *yki*-tumors.

Nevertheless, we understand the importance of this remark and we took the opportunity to rewrite the results section related to role of the insulin signaling in the regulation of *REPTOR* in wildtype flies. In particular, we address that: i) systemic reduction of insulin signaling by *ImpL2* causes a transcriptional upregulation of *REPTOR*, whereas reducing insulin signaling specifically in muscle is not sufficient to raise *REPTOR* expression significantly in thoraces. ii) nutritional restriction has an effect similar to that of *ImpL2*: it increases *REPTOR* expression in thoraces. Our results therefore suggest that in wildtype flies, a decrease in insulin signaling or nutritional restriction, both settings of reduced glucose usage, positively regulates *REPTOR* expression

Together, our data suggests that *ImpL2* produced by the gut modulates *REPTOR* expression via its effects on multiple peripheral tissues, and not simply by reducing insulin signaling specifically in muscle.

3. Experiments demonstrating the links between gut *ImpL2* and the expression levels of *REPTOR* is needed.

In the previous version of the manuscript, we showed that *ImpL2* produced by the gut was necessary and sufficient to increase *REPTOR* expression in thoraces (see new Supplementary Figure 9a, h). Given that we did not detect a significant upregulation of *REPTOR* in thoraces when insulin signaling was reduced in muscle (see new Supplementary Figure 9c), we propose that the transcriptional regulation of *REPTOR* may implicate the effect of *ImpL2* on multiple peripheral tissues. We think that the identification of the factors involved in this potential mechanism of inter-organ communication to control *REPTOR* expression is beyond the scope of the present study.

4. Would *REPTOR* translate to other tumour types? The phenotype appears to be predominant muscle atrophy, maybe not detachment, quantification of thoraces myofiber degradation needs clarification, it is not described neither in figure legend or methods how this is done. How the cut off for counting something as myofiber degradation is determined is not clear.

We will start by answering the first question. There is evidence of tumor models where *ImpL2* is secreted by tumors to induce systemic metabolic effects (PMID: 25850672; PMID: 34473940). Given our hypothesis that *REPTOR* is a repressor of glycolysis, it is possible that in other tumor settings that involve systemic reduction of insulin signaling and likely

decrease in glucose usage by peripheral tissues, REPTOR activity in muscle may be elevated.

Regarding the second comment on scoring the phenotype of muscle degradation, we scanned the dorso-longitudinal muscles of each hemi-thorax looking for the presence of the specific alteration in sarcomere structure that is observed in degraded muscles of either *yki*-flies or when REPTOR activity in muscle is increased (see new Figure 1f, g, new Supplementary Figure 1a-d, new Figure 4d-g, new Supplementary Figure 7b). Occasionally we had to exclude hemi-thoraces in which we were not able confirm the morphology of the sarcomeres across all dorso-longitudinal muscles due to damage caused by dissections.

5. Mechanism into what is activating REPTOR in the muscles in *Yki* tumour animals is not clear.

We have addressed this by showing that the pS6K/S6K ratio is reduced in thoraces of *yki*-flies. These results strongly suggest that the mTOR pathway activity is decreased and therefore the activity of REPTOR should be increased (see new Figure 2b). Importantly, we present new evidence of this negative regulation of REPTOR by mTOR in wildtype muscle tissue. We generated data showing that the co-expression of a *REPTOR*^{WT} allele with PRAS40 or Tsc1/Tsc2, inhibitors of the mTOR pathway, induces the myofiber degradation (see new Figure 4), thereby supporting the hypothesis that the mTOR pathway negatively regulates REPTOR, and its overactivation in muscle leads to myofiber degradation.

6. Some further investigation in the downstream targets of REPTOR, such as pfk, and other members of the metabolic chain, to see if they also rescue would be helpful.

We have addressed this by co-expressing *Myr-AKT*, which increases glycolysis, or knocking-down of *CPT1*, the fatty acid importer to mitochondria: both ameliorated the myofiber degradation induced by *REPTOR* sustained activity. These results strongly support our initial hypothesis that the REPTOR induces the collapse of the muscle fibers by repressing glycolysis, while likely promoting the use of other energy substrates.

Spelling, grammar, clarification

Line 322 "...as 'the' main energy..." – fixed.

Line 357 Saying both REPTOR and CREBRF are functionally conserved makes them sounds like two different things, as opposed to homologues – fixed.

Line 498 Brand name for the plate reader – fixed.

Line 522 Start of this line needs clarification – fixed

Line 531 Brand name for tapestation. – fixed

Line 541 *Drosophila* needs to be in italics – fixed.

Line 547 adjust should be adjusted – fixed.

Line 549 TF should be transcription factor the first time – fixed

Line 609 'read' should be 'red' – fixed

Line 642 Say previous protocols or something, don't just end on 'from' – fixed

Line 659 missing some words at the start of a sentence – fixed

Line 725 why are some plasmids italicised and others are not – fixed.

Line 788 Brand for plate reader – fixed.

Minor Points

Line 97 A description of what was quantified for myofiber degradation would be appreciated. Someone not familiar with what degradation looks like would not understand what you have quantified.

We have rewritten the methods section on myofiber degradation analysis. Also, we have replaced the immunostainings in Figure 1 with new panels that we think illustrate better the degradation of the dorso-longitudinal muscles of *yki*-flies (see new Figure 1d, e). We replaced the immunostainings in Supplementary Figure 1 (see new Supplementary Figure 1a-d) with similar images that include a co-staining with *dlg1* or amphiphysin, which are markers of the t-tubule network (PMID: 11711432), and depict more clearly the degradation of the myofibers. The methods section was changed accordingly to incorporate these new immunostainings.

Line 103 Could this circular morphology be a sign of muscle degradation leading to less pressure on the mitochondria as seen in (state that paper I looked at)

Indeed, and we have changed the text to include this reference. Thank you.

Line 141 A similar length control for the reduction in protein content would be nice

It is unlikely that adult flies do change their size, due to their exoskeleton. As such, we do not think measuring length would be an appropriate control.

Figure 4 and Supp fig 5 Does ATP decrease in flies but increase in the cell model when REPTOR ACT is expressed?

This is a good question, but we did not detect significant changes in ATP content in S2R+ cells expressing *REPTOR^[ACT]*. These results associate with our previous data showing that *REPTOR^[ACT]* did not affect glycolysis in this *in vitro* system. As such, the discrepancies between our *in vivo* and *in vitro* data suggest that S2R+ cells might not be a suitable model to expand our study on the function of REPTOR in regulating energy metabolism in muscle. We have decided to replace the data on metabolic analysis of S2R+ cells with the new *in vivo* data of LC/MS metabolite analysis of thoraces expressing *REPTOR^[ACT]*. The new LC/MS results were included in a new Figure (see new Figure 5) and support our hypothesis that REPTOR acts as a repressor of glycolysis *in vivo*.

Line 523 Should describe the TRIzol extraction 'as above' if that is what it is, otherwise give more details

We have fixed this.

Line 778 Normalised to protein content of each well?

Yes, we have clarified this in the manuscript

Line 799 Should this be listed as a Table?

We have generated a new Supplementary Table 4 listing all qPCR primers used in this work.

The whole RNA extraction and RT-PCR section sounds like it has been written by two different people, as there are inconsistencies in reference placement and tone. I suggest making this more consistent.

We have rewritten this section.

The whole RNA-seq section needs to be checked over for grammar and spelling mistakes.

We apologize for this. We have addressed it.

The drosophila section seems like it needs references, seeing as how many are in the mouse c2c12 cells section

We have addressed this.

Overall there is inconsistency with the use of spaces between values and their respective units. This needs to be addressed throughout

We apologize for this. We have addressed it.

REVIEWER COMMENTS

Reviewer #1 (Remarks to the Author):

The authors have done a great job of addressing the issues that I raised in the initial review, as well as the issues raised by the other reviewers. This study makes a significant contribution to the field and will be of interest to a wide audience.

Reviewer #3 (Remarks to the Author):

We asked about the mechanism of REPTOR, the authors said that they have shown flies with gut yki-tumor have a lower ratio of pS6K/S6K. We are not satisfied with this data. The authors in Figure 2b indeed shows that there is a reduction in this ratio, however, quantification of western blots should be done with n=3 not a single data point. In contrast with their result in 2b, they show in S5E (again only n=1), that there is an increase in the ratio of pS6K/S6K levels. This second data set is done with the LexA-LexAop system, but we do not understand why using this system would generate conflicting data. Thus, it is not possible to conclude that yki-tumors cause altered Tor signalling. The authors should quantify their western blots with n=3. The data generated with the GAL4-UAS system should at least be consistent with that of the LexA-LexAop system.

In point 6 and point 1 of the comments, we suggested the authors try knocking down glycolysis genes downstream of REPTOR. They instead activated upstream mysAKT (which activate many other targets in addition to glycolysis). They also knocked down CPT1, an enzyme in the beta-oxidation pathway, which is not directly involved in glycolysis. They have not addressed whether glycolysis is required downstream of REPTOR. We suggest the authors directly knockdown genes in the glycolysis pathway, and assess if they can ameliorate REPTOR activation phenotype.

REVIEWER COMMENTS

Reviewer #1 (Remarks to the Author):

The authors have done a great job of addressing the issues that I raised in the initial review, as well as the issues raised by the other reviewers. This study makes a significant contribution to the field and will be of interest to a wide audience.

We sincerely thank the reviewer for appreciating this work.

Reviewer #3 (Remarks to the Author):

We asked about the mechanism of REPTOR, the authors said that they have shown flies with gut *yki*-tumor have a lower ratio of pS6K/S6K. We are not satisfied with this data. The authors in Figure 2b indeed shows that there is a reduction in this ratio, however, quantification of western blots should be done with $n=3$ not a single data point. In contrast with their result in 2b, they show in S5E (again only $n=1$), that there is an increase in the ratio of pS6K/S6K levels. This second data set is done with the LexA-LexAop system, but we do not understand why using this system would generate conflicting data. Thus, it is not possible to conclude that *yki*-tumors cause altered Tor signalling. The authors should quantify their western blots with $n=3$. The data generated with the GAL4-UAS system should at least be consistent with that of the LexA-LexAop system.

We would like to clarify that all phosphorylated S6K (pS6K)/ total S6K (tS6K) experiments in question here were replicated at least twice, with experiments performed on different days, using different biological samples. We now make this more clear in our manuscript by including a panel showing two biological replicates with reproducible results for each western blot.

Regarding the inconsistency of the pS6K/ tS6K ratio between the Esg-Gal4 and Esg-LexA systems, we hypothesize that this could be due to a dynamic regulation of the protein levels of tS6K in flies bearing *yki*-gut tumors. We take the opportunity now to expand on this issue.

We detected lower amounts of pS6K in thoraces of *yki*-flies, independent of the system used for *yki* induction in the gut (Esg-Gal4 or Esg-LexA). However, in several of our western blots (as in the case shown in Supp Fig 5e for the Esg-LexA flies), the tS6K levels were also reduced. This causes the pS6K/tS6K ratio to remain similar to that of controls. The interpretation of mTOR activity is not straightforward in the context of changing tS6K levels, owing to the fact that protein degradation may unequally affect the phosphorylated versus non-phosphorylated protein forms of S6K.

Indeed, S6K has been shown to be specifically targeted for ubiquitin-dependent degradation (PMID: 18280803). Importantly, it has been proposed that a second phosphorylation of S6K by JNK kinase activation thorough TNF-alpha, renders the

protein unstable in conditions of decreased mTOR-mediated phosphorylation of S6K (PMID: 23816567). This is a setting analogous to *yki*-tumor flies, which exhibit increased ubiquitination in thoraces, as well as an upregulation of *eiger*, the TNF- α ortholog, in the *yki*-gut tumors (PMID: 34407411). Nevertheless, given the complexity of the regulation of tS6K, to fully understand this point would be beyond the scope of the current work.

ii) Using the Esg-Gal4 system, often samples showed more similar levels of tS6K across experimental conditions. Yet, we also came across *yki*-thorax samples using this expression system that had lower tS6K than control thoraces (See the second replicate in Figure 2b). We speculate that the exact timing of sample harvest may account for variable levels of tS6K levels, even though we tried to tackle this by synchronizing the collection time of samples with a starving/feeding regime. Nevertheless, our results show that in conditions of similar amounts of tS6K levels, *yki*-flies have a lower ratio of pS6K/ tS6K when compared to control flies.

Finally, it is important to reiterate the purpose of these western blots assessing mTOR activity. Our manuscript focuses on the discovery of REPTOR as a regulator of energy metabolism in muscle. We demonstrated its contribution to the muscle metabolic changes observed in *yki*-flies using a loss-of-function model. We then used a gain-of-function model to show that a combination of increased REPTOR protein levels and activation, but not either one alone, is sufficient to induce these same metabolic changes and myofiber degradation. In addition, our qRT-PCR and single nucleus RNA-seq provide evidence that REPTOR expression and activity (as measured by target gene induction) are increased in *yki*-flies, confirming the regulation of REPTOR in muscle in response to *yki*-gut tumors. The S6K experiments above are meant to examine mTOR signaling as one potential REPTOR regulatory input derived from tumors. Our western blots are indeed consistent with the idea that reduced mTOR activity in the context of *yki*-tumors contributes to heightened REPTOR activity, but we do not rule out the idea that other kinases also affect REPTOR activity in this setting. As such, the mTOR western blot results do not affect our manuscript's central conclusions regarding the altered activity and functional contribution of REPTOR in the *yki* gut tumor model.

In point 6 and point 1 of the comments, we suggested the authors try knocking down glycolysis genes downstream of REPTOR. They instead activated upstream *mysAKT* (which activate many other targets in addition to glycolysis). They also knocked down *CPT1*, an enzyme in the beta-oxidation pathway, which is not directly involved in glycolysis. They have not addressed whether glycolysis is required downstream of REPTOR. We suggest the authors directly knockdown genes in the glycolysis pathway, and assess if they can ameliorate REPTOR activation phenotype.

Regarding the comment "We suggest the authors directly knockdown genes in the glycolysis pathway, and assess if they can ameliorate REPTOR activation phenotype", we do not understand this request. We have shown several pieces of evidence that

REPTOR activation represses glycolysis so this experiment does not make sense as it would just reinforce the effect of REPTOR in muscle.

We assume this is a mistake, and that the reviewers are asking us to perform a “rescue” experiment in the REPTOR^[ACT] overexpression background by raising the expression of glycolytic enzymes. While this might seem logical, it raises the problem that the overexpression of a single glycolytic enzyme will likely not be sufficient to rescue the effect of REPTOR on an entire metabolic pathway, especially given our finding that REPTOR reduces several glycolytic metabolites and decreases the expression of multiple genes involved in glycolysis. Furthermore, the overexpression might boost the levels of a single glycolytic enzyme to a point that is not physiological, with unforeseen consequences. We therefore opted to perform the rescue experiment with AKT. This intervention mimics increased insulin signaling, causing a mild upregulation of several glycolytic enzymes (see Supp Fig 2c). When expressed in the muscle of yki-flies or simultaneously with REPTOR^[ACT] in muscle, it ameliorated the myofiber degradation, thereby suggesting that REPTOR sustained activation degrades muscle at least in part by reducing glycolysis. We are aware of the caveats of using AKT and how insulin signaling affects multiple other targets, and we have modified our manuscript text to make this caveat clear. But given the complexity of the system and the technical challenge of altering the expression of multiple glycolytic enzymes simultaneously, we considered this the most reasonable and physiologically appropriate rescue experiment

REVIEWERS' COMMENTS

Reviewer #3 (Remarks to the Author):

I'm satisfied with the revision experiments.